# PrML: Progressive Multi-Task Learning for Monocular 3D Human Pose Estimation

## Abstract

The lifting-based framework has dominated the field of monocular 3D human pose estimation by leveraging the well-detected 2D pose as an intermediate representation. However, it neglects different initial states between 2D pose and per-joint depth. The initial state of the 2D pose is well-detected, but the per-joint depth is unknown and needs to be learned from scratch. The lifting-based framework encodes the well-detected 2D pose and unknown per-joint depth in an entangled feature space, explicitly introducing depth uncertainty to the well-detected 2D pose. To address this limitation, we present a progressive multi-task learning pose estimation framework named **PrML**. First, PrML introduces two task branches to refine the well-detected 2D pose features and to learn the per-joint depth features. This dual-branch design reduces the explicit influence of uncertain depth features on 2D pose features. Second, PrML employs a task-aware decoder to indirectly supplement the complementary information between the refined 2D pose features and learned per-joint depth features. This step establishes the connection between 2D pose and per-joint depth, compensating for the lack of interaction caused by the dual-branch design. We conduct theoretical analysis from the perspective of mutual information and arrive at a loss to supervise this feature complementary process. Finally, we use two regression heads to regress the 2D pose and per-joint depth, respectively, and concatenate them to obtain the final 3D pose. Extensive experiments show that PrML outperforms the conventional lifting-based framework with fewer parameters on two widely used datasets: Human3.6M and MPI-INF-3DHP. Code is available at https://anonymous.4open.science/r/PrML.

## 1 Introduction

Monocular 3D human pose estimation has been a crucial problem in computer vision, which aims to locate the 3D joint positions of a human body (Moon & Lee, 2020; Pavlakos et al., 2018; Chen et al., 2021). Nowadays, monocular 3D human pose estimation finds widespread applications in various scenarios, including motion prediction (Liu et al., 2021b; 2022b), action recognition (Zhang et al., 2022a), and human-robot interaction (Gong et al., 2022; Ye et al., 2021). Existing monocular 3D human pose estimation methods can be categorized as the end-to-end manner and lifting-based manner. The end-to-end approaches (Kanazawa et al., 2018; Pavlakos et al., 2017; Sun et al., 2018) directly estimate the 3D pose from the input image without the intermediate 2D pose representation. Different from the end-to-end manner, lifting-based methods (Martinez et al., 2017; Liu et al., 2020) first obtain 2D pose using 2D pose detector (Newell et al., 2016; Chen et al., 2018) and then lift the 2D pose in image coordinate to the 3D pose in camera coordinate. These lifting-based methods usually outperform the end-to-end manner and dominate the monocular 3D human pose estimation.

Recent lifting-based methods (Zheng et al., 2021; Li et al., 2022b; Zhang et al., 2022b; Yu et al., 2023; Zhu et al., 2023; Peng et al., 2024) for monocular 3D human pose estimation focus on designing various spatio-temporal encoders. As shown in Figure 1 left, they project the 2D pose into an entangled feature space and regress the 3D pose from it. This lifting process neglects the different initial states of 2D pose and per-joint depth. **It encodes the well-detected 2D pose and unknown per-joint depth in an entangled feature space, which introduces a main limitation: the high uncertainty of the per-joint depth may erode the 2D pose.** It is well-known that the monocular 3D human pose estimation task is an ill-posed problem and inherently suffers from depth ambiguity (Li et al., 2022b; Ma et al., 2021b; Wehrbein et al., 2021). One 2D pose possibly corresponds to multiple

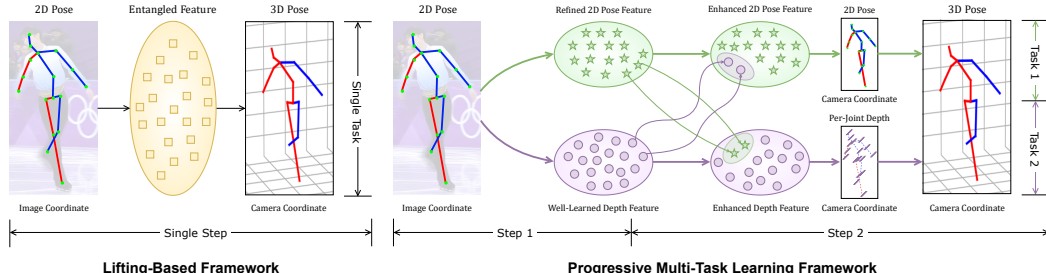

Figure 1: Given a 2D pose in the image coordinate, we aim to estimate the 3D pose in the camera coordinate. Left: Conventional lifting-based framework directly projects the 2D pose in an entangled feature space and regression the 3D pose from it. Right: Our proposed progressive multi-task learning framework. The 2D pose and per-joint depth features are learned separately in the first step. In the second step, we perform feature interaction to supplement the complementary information. Finally, we regress the 2D pose and per-joint depth and concatenate them to obtain the final 3D pose.

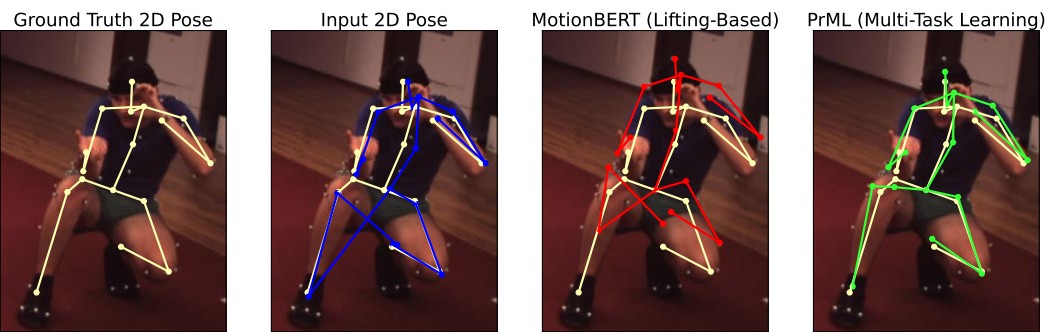

Figure 2: Qualitative Comparison of 2D Pose (Ground Truth, Input, MotionBERT (Zhu et al., 2023) and Ours). We project the 2D pose in the camera coordinate (part of the output 3D pose) back to the image coordinate for comparison. The powerful lifting-based method MotionBERT gets a 2D pose worse than the input, which contradicts our intuition. In contrast, our framework obtains a 2D pose better than the input. Please refer to Appendix F for more qualitative and quantitative comparisons.

3D poses, where the lifting process is inherently ambiguous (Yu et al., 2021). To validate the impact of depth uncertainty on the 2D pose, we project the 2D pose in the camera coordinate (part of output 3D pose) back to the image coordinate and compare it with the ground truth 2D pose and input 2D pose. As shown in Figure 2, despite learning through multiple spatio-temporal encoders, the 2D pose of the powerful lifting method MotionBERT (Zhu et al., 2023) is even worse than the original input 2D pose. This observation provides empirical evidence that directly encoding the well-detected 2D pose features and the unknown per-joint depth features in an entangled feature space will inevitably introduce explicit uncertainty to the 2D pose and cause erosion. To provide more empirical support for the high uncertainty of per-joint depth, we conduct quantitative comparisons of Mean Per Joint Position Error (MPJPE) across different axes for different hard actions (Zeng et al., 2021) with MotionBERT (Zhu et al., 2023), GLA-GCN (Yu et al., 2023) and KTPFormer (Peng et al., 2024). As shown in Figure 3, the MPJPE of per-joint depth is significantly higher than the MPJPE of 2D pose and accounts for the majority of the overall MPJPE. These quantitative findings highlight the high uncertainty of per-joint depth compared to the well-detected 2D pose.

Motivated by these qualitative and quantitative observations, we propose a progressive multi-task learning pose estimation framework named **PrML** to address this limitation. As shown in Figure 1 right, the first step of PrML introduces two task branches: refining the well-detected 2D pose features and learning the per-joint depth features. The dual-branch design brings two benefits. First, learning the features of the 2D pose and per-joint depth separately avoids the explicit impact of uncertain depth features on the 2D pose. Second, the model parameters are not shared across two task branches, which makes the training more targeted. After dual-branch learning, we obtain the refined 2D pose features and learned depth features, mitigating the uncertainty of depth features. In light of this, the second

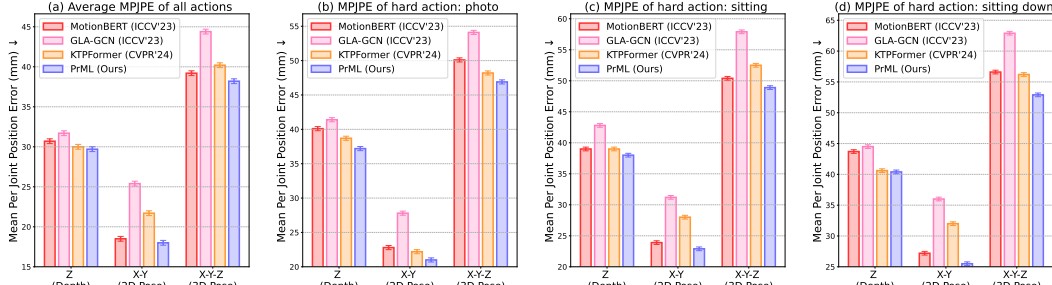

Figure 3: Quantitative Comparison of Mean Per Joint Position Error (MPJPE) of different axes for all actions and three hard actions (Zeng et al., 2021) with lifting-based methods (Zhu et al., 2023; Yu et al., 2023; Peng et al., 2024). The MPJPE of the Z-axis (per-joint depth) is significantly higher than the X-Y axes (2D pose) and accounts for the majority of the overall (3D pose) MPJPE. Our proposed framework achieves better results across different axes than the lifting-based framework.

step of PrML employs a task-aware decoder to indirectly supplement the complementary information between the refined 2D pose features and the learned per-joint depth features. This step compensates for the information loss caused by the dual-branch structure and establishes the connection between 2D pose and per-joint depth. We also conduct theoretical analysis from the perspective of mutual information (Becker, 1996) and arrive at a loss to supervise this feature complementary process. Finally, we regress the 2D pose and per-joint depth, respectively, and concatenate them to obtain the final 3D pose. As shown in Figure 2, our framework could reduce the erosion of the 2D pose caused by depth uncertainty. The quantitative results in Figure 3 also demonstrate our framework performs favorably across different parts (2D pose and per-joint depth) of the 3D pose. Extensive experiments on two widely used monocular 3D human pose estimation benchmarks (i.e., Human3.6M (Ionescu et al., 2013) and MPI-INF-3DHP (Mehta et al., 2017)) demonstrate that the proposed progressive multi-task learning framework outperforms conventional lifting-based framework in terms of accuracy and robustness with fewer parameters. The key contributions of this paper are as follows:

- We tackle an overlooked different initial states between the well-detected 2D pose and the unknown per-joint depth of the lifting-based framework and present a novel progressive multi-task learning pose estimation framework named PrML to address it.
- We propose a task-aware decoder to indirectly supplement the complementary information between 2D pose and per-joint depth after task learning. We also conduct theoretical analysis from the perspective of mutual information to explicitly supervise this feature complementary process.
- Our framework achieves state-of-the-art results on Human3.6M and MPI-INF-3DHP datasets with fewer parameters. These results demonstrate the potential of the progressive multi-task learning framework for future monocular 3D human pose estimation research.

## 2 RELATED WORK

**Monocular 3D Human Pose Estimation.** Existing methods for monocular 3D human pose estimation can be categorized as end-to-end and lifting-based. End-to-end approaches (Kanazawa et al., 2018; Pavlakos et al., 2017; Sun et al., 2018) directly estimate the 3D pose from the input image without the intermediate 2D pose representation. With the reliable achievement of 2D human pose detectors (Chen et al., 2018; Newell et al., 2016; Sun et al., 2019), lifting-based methods (Fang et al., 2018; Martinez et al., 2017; Zhao et al., 2019; Liu et al., 2020) first obtain 2D pose representations in the image and then lift the 2D joint coordinates to 3D space. Recently, Transformers (Vaswani et al., 2017) have been applied to various visual tasks (Dosovitskiy et al., 2021; Carion et al., 2020). For the monocular 3D human pose estimation task, PoseFormer (Zheng et al., 2021) introduces transformer architecture to leverage spatial and temporal dependency. MHFormer (Li et al., 2022b) addresses the depth ambiguity by learning multiple pose hypotheses and MixSTE (Zhang et al., 2022b) constructs a mixed spatiotemporal transformer to capture the temporal motion of different body joints. STCFormer (Tang et al., 2023) decomposed spatio-temporal attention and integrated the structure-enhanced positional embedding. On the other hand, MotionBERT (Zhu et al., 2023) trains

a unified model for multiple downstream tasks. In (Peng et al., 2024), KTPFormer uses two prior attention modules to facilitate pose estimation. Moreover, MotionAGFormer (Soroush Mehraban, 2024) using two parallel transformer and GCNFormer streams to better learn the underlying 3D structure. However, these methods are developed within the conventional lifting-based framework. In contrast, we propose a progressive multi-task learning framework to estimate 3D human pose.

**Multi-Task Learning.** Multi-Task Learning (MTL) (Caruana, 1997) is a learning paradigm in machine learning, and it aims to leverage useful information contained in multiple related tasks to help improve the generalization performance of all the tasks (Zhang & Yang, 2021). Numerous models have been explored (Vandenhende et al., 2020; Brüggemann et al., 2021) within the MTL framework. Moreover, existing approaches analyze the optimization of multi-task learning by designing multi-task loss (Liu et al., 2021a; Li et al., 2022a) or gradient manipulations (Yu et al., 2020; Wang et al., 2020). MTL has been widely used in computer vision, such as image classification (Rebuffi et al., 2017), semantic segmentation (Hoyer et al., 2021; Li et al., 2023), and dense prediction (Proesmans et al., 2022; Hoyer et al., 2021). In (Iqbal et al., 2018), they introduce a novel scale and translation invariant 2.5D pose representation contain 2D pose and depth. Our approach is motivated by these former attempts but from the perspective of decomposing the single 3D human pose estimation task into two sub-tasks and learning them in a progressive manner, which is a novel and unexplored question.

**Mutual Information.** Mutual Information plays an important role in the representation learning. As the pioneering work among mutual information methods, Linsker (Linsker, 1988) proposes to maximize mutual information between the input and output. Designing optimization objectives based on mutual information maximization has been extensively studied (Becker, 1992; Wiskott & Sejnowski, 2002). For human pose estimation, CV-MIM (Zhao et al., 2021) introduces a representation learning method to disentangle pose-dependent and view-dependent factors from 2D human poses. FAMI-Pose (Liu et al., 2022a) designs a mutual information loss to maximize the complementary information between temporal frames. TDMI (Feng et al., 2023) proposes to minimize the mutual information between useful and noisy constituents of the raw features. To the best of our knowledge, we are the first to introduce mutual information loss to the monocular 3D human pose estimation task.

## 3 RETHINKING LIFTING-BASED MONOCULAR 3D HUMAN POSE ESTIMATION

Since SimpleBaseline (Martinez et al., 2017) proposes the 2D-to-3D lifting framework, numerous methods (Pavllo et al., 2019; Zhang et al., 2022b; Li et al., 2022b; Shan et al., 2022; Zhao et al., 2023b; Shan et al., 2023; Zhu et al., 2023; Tang et al., 2023; Peng et al., 2024; Mehraban et al., 2024) have been developed within this framework. These lifting-based methods usually outperform the end-to-end manner (Kanazawa et al., 2018; Pavlakos et al., 2017; Sun et al., 2018) and have been the dominant paradigm in monocular 3D human pose estimation for a long time.

The ensuing question is why lifting-based methods perform better than end-to-end approaches. We argue that this is mainly attributed to leveraging the 2D pose as an intermediate representation. First, there exists a high relevance between 2D pose and 3D pose. Regressing 3D pose directly from raw images is a highly nonlinear and challenging problem (Pavlakos et al., 2017). This difficulty also exists in 2D human pose estimation (Pfister et al., 2015; Tompson et al., 2014). In contrast, with the widespread usage of 2D human pose detectors (Chen et al., 2018; He et al., 2017; Newell et al., 2016; Sun et al., 2019), lifting-based methods could leverage the well-detected 2D pose, which contributes to its 3D counterpart and make network training easy. Second, the 2D pose is exceptionally lightweight regarding memory cost compared to raw image. This property enables lifting-based methods to leverage long-term temporal clues to address the occlusion and achieve advanced accuracy. (e.g., 243 frames for MixSTE (Zhang et al., 2022b), MotionBERT (Zhu et al., 2023), and KTPFormer (Peng et al., 2024); large as 351 frames for MHFormer (Li et al., 2022b))

Once we have a well-detected 2D pose, lifting it directly to 3D space is natural and simple. However, these lifting-based methods neglect different initial states between 2D pose and per-joint depth and encode the well-detected 2D pose features and unknown per-joint depth features in an entangled feature space. This leads to the fact that despite these methods (Zhu et al., 2023; Peng et al., 2024; Li et al., 2022b) striving to design various encoders to leverage the well-detected 2D pose, the 2D pose itself is inevitably eroded by the uncertainty of depth features (see Figure 2). This paper presents a progressive multi-task learning framework that addresses the different initial states between 2D pose and per-joint depth and provides a new choice for future monocular 3D human pose estimation.

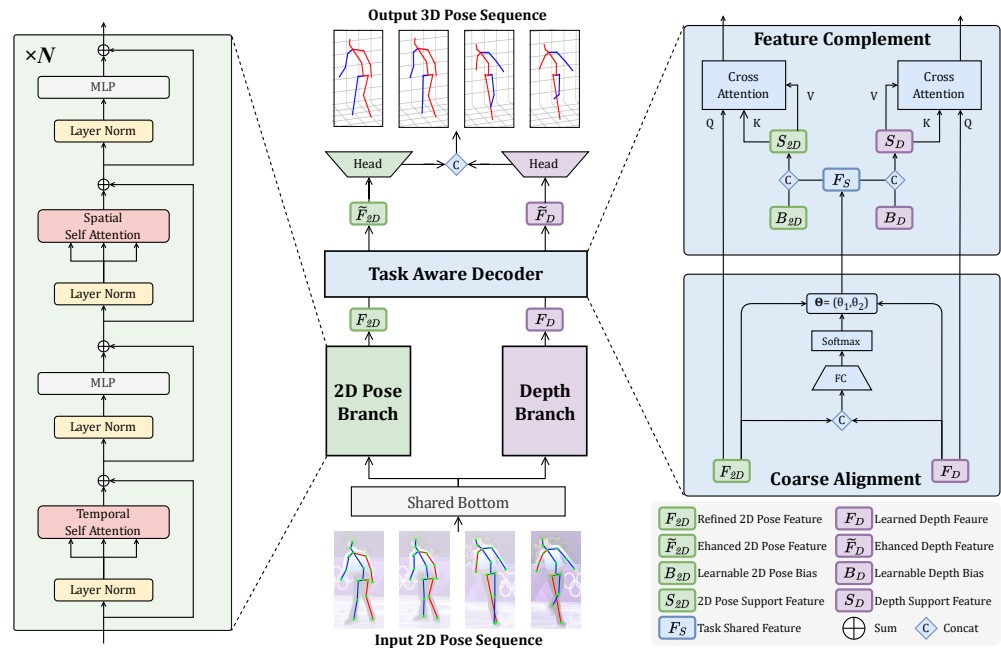

Figure 4: Overview of the proposed progressive multi-task learning framework PrML, which comprises a shared bottom, a 2D pose branch, a depth branch, and a task-aware decoder.

# 4 METHODOLOGY

## 4.1 PROBLEM FORMULATION

Given a 2D pose sequence $X \in \mathbb{R}^{T \times J \times C_{\text{in}}}$, the goal of monocular 3D human pose estimation is to estimate the 3D pose sequence $Y \in \mathbb{R}^{T \times J \times C_{\text{out}}}$. Here, $T$ refers to the number of input frames, and $J$ refers to the number of joints. $C_{\text{in}}$ and $C_{\text{out}}$ denote the dimension of the input and output.

## 4.2 MULTI-TASK LEARNING BRANCH

One of the widely used multi-task learning models is proposed by Caruana (Caruana, 1997; 1993), which has a shared-bottom model structure that substantially reduces the risk of overfitting (Ma et al., 2018). We first use a linear embedding layer to project the 2D pose sequence into high-dimensional features. Then, we employ the DSTFormer block proposed by MotionBERT (Zhu et al., 2023) as our shared bottom to extract general features $F \in \mathbb{R}^{T \times J \times C}$. The DSTFormer block is composed of spatial-temporal and temporal-spatial branches. The outputs of two branches are adaptively fused by an attention regressor. Next, we add the learnable 2D pose position embedding and per-joint depth position embedding to $F$ to obtain the 2D pose features $F_{2D} \in \mathbb{R}^{T \times J \times C}$ and the per-joint depth features $F_D \in \mathbb{R}^{T \times J \times C}$ respectively. $C$ denotes the feature dimension. Subsequently, the 2D pose branch and depth branch repeat the temporal transformer encoder ($TF_T$) and spatial transformer encoder ($TF_S$) for $N$ times to refine the 2D pose features and learn depth features separately as:

$$F_{2D}^n = TF_S(TF_T(F_{2D}^{n-1})) \quad F_D^n = TF_S(TF_T(F_D^{n-1})) \qquad n = 1 \dots N \qquad (1)$$

## 4.3 TASK-AWARE DECODER

**Coarse Alignment.** We first use a fully connected layer $\mathcal{F}_{FC}(\cdot)$ and softmax function $Softmax(\cdot)$ to compute the coarse alignment parameters $\Theta = (\theta_1, \theta_2) \in \mathbb{R}^{T \times J \times 2}$ to project the 2D pose features and per-joint depth features into a shared feature space and obtain the shared features $F_S \in \mathbb{R}^{T \times J \times C}$. This coarse alignment operation mitigates the uncertainty erosion by avoiding direct interaction

between the 2D pose feature and the per-joint depth feature and can be expressed as follows:

$$F_S = \underbrace{Softmax(\mathcal{F}_{FC}(F_{2D}^N \oplus F_D^N))}_{\Theta} \begin{pmatrix} F_{2D}^N \\ F_D^N \end{pmatrix} = \theta_1 F_{2D}^N + \theta_2 F_D^N \tag{2}$$

**Feature Complement.** The shared features $F_S$ obtained after coarse alignment have the 2D pose and per-joint depth features but lack precise and targeted support information. Thus, we introduce the learnable 2D pose bias $B_{2D} \in \mathbb{R}^{T \times J \times C}$ and per-joint depth bias $B_D \in \mathbb{R}^{T \times J \times C}$ to address this issue. We concatenate the task bias $B_{2D}$ and $B_D$ with the shared features $F_S$ to obtain the 2D pose support features $S_{2D} \in \mathbb{R}^{T \times J \times 2C}$ and depth support features $S_D \in \mathbb{R}^{T \times J \times 2C}$. Then, we utilize Multi-Head Cross-Attention (Vaswani et al., 2017) ($TF_C$) with the original features acting as the query and the support features serving as the key and value for feature supplementation. Technically, we get the enhanced 2D pose features $\widetilde{F}_{2D}$ and enhanced depth features $\widetilde{F}_D$ as follows:

$$\widetilde{F}_{2D} = TF_C(F_{2D}^N, S_{2D}) \quad \widetilde{F}_D = TF_C(F_D^N, S_D) \tag{3}$$

### 4.4 MUTUAL INFORMATION OBJECTIVE

**Mutual Information.** Mutual information (MI) is an important measurement to quantify the statistical dependency of two random variables. Given two random variables $x$ and $y$, $p(x, y)$ represents the joint probability distribution between $x$ and $y$, while $p(x)$ and $p(y)$ represent their marginal distributions. The mutual information between two random variables $x$ and $y$ is defined as:

$$\mathcal{I}(X;Y) = \int_Y \int_X p(x,y) \log \left( \frac{p(x,y)}{p(x)p(y)} \right) dxdy \tag{4}$$

**Mutual Information Loss.** Within the task-aware decoder, our goal is to explicitly supervise the feature complementary process. This mutual information objective can be formulated as follows:

$$\max \left[ \mathcal{I}(Y_{2D}; S_{2D} \mid B_{2D}) + \mathcal{I}(Y_D; S_D \mid B_D) \right] \tag{5}$$

where $Y_{2D}$ and $Y_D$ denote the 2D pose and per-joint depth of the 3D pose label. Intuitively, optimizing this objective will maximize the mutual information between the support feature and the label to ehance the feature complementary. Due to the notorious difficulty of the conditional MI computations especially in neural networks (Hjelm et al., 2018; Tian et al., 2021), we factorize Equation 5 as:

$$\mathcal{I}(Y_D; S_D \mid B_D) = \mathcal{I}(Y_D; S_D) - \mathcal{I}(S_D; B_D) + \int_{Y_D} \underbrace{D_{KL}(P_{(S_D, B_D)|Y_D} \| P_{S_D|Y_D} P_{B_D|Y_D})}_{\text{KL Divergence} \geq 0} dP_{Y_D}$$

$$\geq \mathcal{I}(Y_D; S_D) - \mathcal{I}(S_D; B_D) \tag{6}$$

Since both $\mathcal{I}(Y_D; S_D)$ and $\mathcal{I}(S_D; B_D)$ are non-negative, the $\mathcal{I}(Y_D; S_D) - \mathcal{I}(S_D; B_D)$ will result in negative values during training. This will yield negative values during training, leading to vanishing gradients problem and preventing training from converging. Therefore, we simplified the implementation of mutual information by calculating only the first term $\mathcal{I}(Y_D; S_D)$. Finally, we obtain two simplified mutual information optimization objectives as follows:

$$\mathcal{L}_{MI} = \lambda_{2D}\mathcal{I}(Y_{2D}; S_{2D}) + \lambda_D \mathcal{I}(Y_D; S_D) \tag{7}$$

The $\lambda_{2D}$ and $\lambda_D$ serve as hyper-parameters in our framework to balance different objects.

### 4.5 REGRESSION HEAD AND LOSS FUNCTION

We use two regression heads (MLP) to regress the 2D pose $\overline{Y}_{2D} \in \mathbb{R}^{T \times J \times 2}$ and per-joint depth $\overline{Y}_D \in \mathbb{R}^{T \times J \times 1}$ respectively and concatenate them to generate the 3D pose sequence $\overline{Y}$. Losses are independently calculated for 2D pose and per-joint depth as Equation 8. For the 2D pose, we use L2 loss to minimize the errors between predictions and ground truth. For the per-joint depth, we use mean absolute error loss to minimize the errors between the estimated per-joint depth and label.

$$\mathcal{L}_{3D} = \underbrace{\frac{1}{JT} \sum_{j=1}^{J} \sum_{t=1}^{T} \left\| Y_{2D}^{j,t} - \overline{Y}_{2D}^{j,t} \right\|_2}_{\text{2D Pose Optimization Objective}} + \underbrace{\frac{1}{JT} \sum_{j=1}^{J} \sum_{t=1}^{T} \left| Y_D^{j,t} - \overline{Y}_D^{j,t} \right|}_{\text{Depth Optimization Objective}} \tag{8}$$

Where $Y_{2D}^{j,t}$ and $Y_D^{j,t}$ are the 2D pose and per-joint depth of 3D pose label. $\overline{Y}_{2D}^{j,t}$ and $\overline{Y}_D^{j,t}$ are the predicted results of the $j$-th joint in $t$-th frame. In addition, the temporal consistency loss $\mathcal{L}_T$ from (Hossain & Little, 2018) is introduced to produce smooth poses. The total loss $\mathcal{L}$ is defined as follows:

$$\mathcal{L} = \mathcal{L}_{3D} + \lambda_T \mathcal{L}_T + \lambda_{MI} \mathcal{L}_{MI} \tag{9}$$

where $\lambda_T$ and $\lambda_{MI}$ are hyper-parameters to balance the ratio of different loss terms.

# 5 EXPERIMENTS

## 5.1 EXPERIMENT SETTING

We evaluate our model on two large-scale monocular 3D human pose estimation datasets: Human3.6M (Ionescu et al., 2013) and MPI-INF-3DHP (Mehta et al., 2017). For the Human3.6M dataset, we report the MPJPE (Mean Per Joint Position Error) and P-MPJPE (Procrustes-MPJPE) as evaluation metrics as prior methods (Li et al., 2022b; Zhu et al., 2023; Zhang et al., 2022b; Zhao et al., 2023b). For the MPI-INF-3DHP dataset, similar to existing approaches (Shan et al., 2022; Tang et al., 2023; Chen et al., 2023; Zhu et al., 2023), we use ground truth 2D pose as input and report MPJPE, Percentage of Correct Keypoint (PCK) with the threshold of 150mm, and Area Under Curve (AUC) as the evaluation metrics. Please refer to Appendix B for implementation details.

Table 1: Results on Human3.6M in millimeters (mm) under MPJPE using 2D pose detected by SH (Newell et al., 2016) following MotionBERT (Zhu et al., 2023). T is the length of the input 2D pose sequence. The best result is shown in bold, and the second-best result is underlined.

| MPJPE | T | Dir. | Disc. | Eat | Greet | Phone | Photo | Pose | Pur. | Sit | SitD. | Smoke | Wait | WalkD. | Walk | WalkT. | Avg |
|---|---|---|---|---|---|---|---|---|---|---|---|---|---|---|---|---|---|
| MHFormer (Li et al., 2022b) | 81 | - | - | - | - | - | - | - | - | - | - | - | - | - | - | - | 44.5 |
| MixSTE (Zhang et al., 2022b) | 81 | 39.8 | 43.0 | 38.6 | 40.1 | 43.4 | 50.6 | 40.6 | 41.4 | 52.2 | 56.7 | 43.8 | 40.8 | 43.9 | 29.4 | 30.3 | 42.4 |
| P-STMO (Shan et al., 2022) | 81 | 41.7 | 44.5 | 41.0 | 42.9 | 46.0 | 51.3 | 42.8 | 41.3 | 54.9 | 61.8 | 45.1 | 42.8 | 43.8 | 30.8 | 30.7 | 44.1 |
| PoseFormerV2 (Zhao et al., 2023b) | 81 | - | - | - | - | - | - | - | - | - | - | - | - | - | - | - | 46.0 |
| STCFormer (Tang et al., 2023) | 81 | 40.6 | 43.0 | 38.3 | 40.2 | 43.5 | 52.6 | 40.3 | 40.1 | 51.8 | 57.7 | **42.8** | 39.8 | 42.3 | **28.0** | 29.5 | 42.0 |
| GLA-GCN (Yu et al., 2023) | 81 | - | - | - | - | - | - | - | - | - | - | - | - | - | - | - | - |
| MotionBERT (Zhu et al., 2023) | 81 | - | - | - | - | - | - | - | - | - | - | - | - | - | - | - | - |
| KTPFormer (Peng et al., 2024) | 81 | **39.1** | 41.9 | **37.3** | 40.1 | 44.0 | 51.3 | **39.8** | 41.0 | 51.4 | 56.0 | 43.0 | 41.0 | 42.6 | 28.8 | 29.5 | 41.8 |
| MotionAGFormer (Soroush Mehraban, 2024) | 81 | 41.9 | 42.7 | 40.4 | 37.6 | 45.6 | 51.3 | 41.0 | 38.0 | 54.1 | 58.8 | 45.5 | 40.4 | 39.8 | 29.4 | 31.0 | 42.5 |
| **PrML (Ours)** | 81 | 39.7 | **41.4** | 39.4 | **35.5** | **43.1** | 50.7 | 40.0 | 37.2 | 51.1 | 56.0 | 43.7 | 40.5 | **39.1** | 28.7 | **28.8** | **41.0** |
| MHFormer (Li et al., 2022b) | 351 | 39.2 | 43.1 | 40.1 | 40.9 | 44.9 | 51.2 | 40.6 | 41.3 | 53.5 | 60.3 | 43.7 | 41.1 | 43.8 | 29.8 | 30.6 | 43.0 |
| MixSTE (Zhang et al., 2022b) | 243 | 37.6 | 40.9 | **37.3** | 39.7 | 42.3 | 49.9 | 40.1 | 39.8 | 51.7 | 55.0 | 42.1 | 39.8 | 41.0 | 27.9 | 27.9 | 40.9 |
| P-STMO (Shan et al., 2022) | 243 | 38.9 | 42.7 | 40.4 | 41.1 | 45.6 | 49.7 | 40.9 | 39.9 | 55.5 | 59.4 | 44.9 | 42.2 | 42.7 | 29.4 | 29.4 | 42.8 |
| PoseFormerV2 (Zhao et al., 2023b) | 243 | - | - | - | - | - | - | - | - | - | - | - | - | - | - | - | 45.2 |
| STCFormer (Tang et al., 2023) | 243 | 39.6 | 41.6 | 37.4 | 38.8 | 43.1 | 51.1 | 39.1 | 39.7 | 51.4 | 57.4 | 41.8 | 38.5 | 40.7 | 27.1 | 28.6 | 41.0 |
| GLA-GCN (Yu et al., 2023) | 243 | 41.3 | 44.3 | 40.8 | 41.8 | 45.9 | 54.1 | 42.1 | 41.5 | 57.8 | 62.9 | 45.0 | 42.8 | 45.9 | 29.4 | 29.9 | 44.4 |
| MotionBERT (Zhu et al., 2023) | 243 | 36.3 | 38.7 | 38.6 | 33.6 | 42.1 | 50.1 | **36.2** | 35.7 | 50.1 | 56.6 | 41.3 | 37.4 | 37.7 | **25.6** | 26.5 | 39.2 |
| KTPFormer (Peng et al., 2024) | 243 | 37.3 | 39.2 | **35.9** | 37.6 | 42.5 | 48.2 | 38.6 | 39.0 | 51.4 | 55.9 | 41.6 | 39.0 | 40.0 | 27.0 | 27.4 | 40.1 |
| MotionAGFormer (Soroush Mehraban, 2024) | 243 | 36.8 | 38.5 | **35.9** | 33.0 | 41.1 | 48.6 | 38.0 | 34.8 | 49.0 | 51.4 | 40.3 | 37.4 | 36.3 | 27.2 | 27.2 | 38.4 |
| **PrML (Ours)** | 243 | **36.0** | **38.2** | 37.3 | 33.5 | 40.4 | 46.9 | 37.5 | **34.6** | 48.9 | 52.9 | 40.7 | 36.6 | 36.7 | 26.1 | **26.1** | **38.2** |

Table 2: Results on Human3.6M in millimeters (mm) under MPJPE using ground truth 2D pose. T is the number of input frames. Seq2seq refers to estimating 3D pose sequences rather than only the center frame. MACs/frames represents multiply-accumulate operations for each output frame. The best result is shown in bold, and the second-best result is underlined.

| Method | Venue | Framework | Seq2Seq | T | Parameter | MACs | MACs/frame | MPJPE |
|---|---|---|---|---|---|---|---|---|
| MHFormer (Li et al., 2022b) | CVPR'22 | Lifting-Based | ✗ | 351 | 30.9M | 7.1G | 7096M | 30.5 |
| MixSTE (Zhang et al., 2022b) | CVPR'22 | Lifting-Based | ✓ | 243 | 33.6M | 139.0G | 572M | 21.6 |
| P-STMO (Shan et al., 2022) | ECCV'22 | Lifting-Based | ✗ | 243 | 6.2M | 0.7G | 740M | 29.3 |
| PoseFormerV2 (Zhao et al., 2023b) | CVPR'23 | Lifting-Based | ✗ | 243 | 14.3M | 0.5G | 528M | - |
| STCFormer (Tang et al., 2023) | CVPR'23 | Lifting-Based | ✓ | 243 | 4.7M | 19.6G | 80M | 21.3 |
| GLA-GCN (Yu et al., 2023) | ICCV'23 | Lifting-Based | ✗ | 243 | 1.3M | 1.5G | 1556M | 21.0 |
| MotionBERT (Zhu et al., 2023) | ICCV'23 | Lifting-Based | ✓ | 243 | 42.5M | 174.7G | 719M | 17.8 |
| KTPFormer (Peng et al., 2024) | CVPR'24 | Lifting-Based | ✓ | 243 | 33.7M | 69.5G | 286M | 19.0 |
| MotionAGFormer (Soroush Mehraban, 2024) | WACV'24 | Lifting-Based | ✓ | 243 | 19.0M | 78.3G | 322M | 17.3 |
| **PrML (Ours)** | - | Multi-Task Learning | ✓ | 243 | 13.0M | 49.3G | 203M | **17.2** |

## 5.2 COMPARISON WITH STATE-OF-THE-ART METHODS

**Human3.6M.** We compare our method with several state-of-the-art techniques on the Human3.6M dataset. For fair comparisons, only the results of models without extra pre-training on additional data are included. Table 1 summarizes the performance comparisons in terms of MPJPE of all 15 actions, and the number of the input frames T is also given for each method. Our method achieved state-of-the-art performance with an MPJPE of 38.2mm with T = 243. It is worth noting that our method in the case of T = 81 input frames still achieves state-of-the-art performance with an MPJPE

error of 41.0mm and even surpasses the performance of several methods with a higher number of input frames. For example, this result outperforms P-STMO (Shan et al., 2022) (41.0mm v.s. 42.8mm), PoseformerV2 (Zhao et al., 2023b) (41.0mm v.s. 45.2mm) with 243 frames, and MHFormer (Li et al., 2022b) even with 351 frames (41.0mm v.s. 43.0mm). These results demonstrate the effectiveness of PrML. To further validate the effectiveness of the multi-task learning framework, we also report the model parameters, MACs (Multiply–Accumulate Operations), and MPJPE using 2D ground truth as input. As shown in Table 2, our method with T = 243 achieves the best performance with an MPJPE of 17.2mm, which outperforms the lifting-based framework with faster inference speed. For example, this result outperforms MotionBERT (Zhu et al., 2023) (17.2mm v.s. 17.8mm). Due to space limitations, we present the results of P-MPJPE (Procrustes-MPJPE) in Appendix C.

**MPI-INF-3DHP.** To demonstrate the generalization capability of our model, we also evaluate our model on the challenging MPI-INF-3DHP dataset, which includes more complex scenes and motions. Following previous works (Zheng et al., 2021; Zhang et al., 2022b; Shan et al., 2022; Tang et al., 2023; Li et al., 2022b; 2024), we use ground truth 2D pose as input and set the number of input frames as 9, 27, or 81. As observed in Table 3, our method with T = 81 achieves the best performance with

Table 3: Results on MPI-INF-3DHP dataset under PCK, AUC, and MPJPE using ground truth 2D pose as input. T is the number of input frames. Seq2seq refers to estimating 3D pose sequence. (*) indicate our re-implementation.

| Method | T | Seq2Seq | PCK↑ | AUC↑ | MPJPE↓ |
|---|---|---|---|---|---|
| MHFormer (Li et al., 2022b) | 9 | ✗ | 93.8 | 63.3 | 58.0 |
| MixSTE (Zhang et al., 2022b) | 27 | ✓ | 94.4 | 66.5 | 54.9 |
| P-STMO (Shan et al., 2022) | 81 | ✗ | 97.9 | 75.8 | 32.2 |
| PoseFormerV2 (Zhao et al., 2023b) | 81 | ✗ | 97.9 | 78.8 | 27.8 |
| GLA-GCN (Yu et al., 2023) | 81 | ✗ | 98.5 | 79.1 | 27.8 |
| STCFormer (Tang et al., 2023) | 81 | ✓ | 98.7 | 83.9 | 23.1 |
| MotionBERT* (Zhu et al., 2023) | 81 | ✓ | 98.7 | 85.6 | 16.5 |
| KTPFormer (Peng et al., 2024) | 81 | ✓ | **98.9** | 85.9 | 16.7 |
| MotionAGFormer (Soroush Mehraban, 2024) | 81 | ✓ | 98.2 | 85.3 | 16.2 |
| **PrML** (Ours) | 9 | ✓ | 98.1 | 82.2 | 23.3 |
| **PrML** (Ours) | 27 | ✓ | 98.6 | 85.8 | 18.1 |
| **PrML** (Ours) | 81 | ✓ | **98.9** | **86.9** | **15.7** |

the PCK of 98.9%, AUC of 86.9%, and MPJPE of 15.7mm. Similar to the previous findings, our method with T = 9, 27 input frames still outperforms the previous state-of-the-art methods and achieves the MPJPE of 23.3mm and 18.1mm, respectively. More remarkably, our method with T = 9 input frames outperforms the GLA-GCN (Yu et al., 2023) with T = 81 input frames, despite having only one-ninth of the input frames (23.3mm v.s. 27.8mm, 9 frames vs. 81 frames).

**Robustness to Noisy 2D Pose.** Benefiting from the 2D pose branch, our method not only preserves well-detected 2D pose features but also allows us to handle noisy 2D pose input. To demonstrate that the inclusion of the 2D pose branch helps improve the robustness of the proposed method, we make the pose estimation task more challenging by adding zero-mean Gaussian noise to the ground-truth 2D pose on the Human3.6M (Ionescu et al., 2013) and MPI-INF-3DHP (Mehta et al., 2017). As shown in Figure 5, the experimental evidence reveals that our proposed PrML suffers from less performance drop as the standard deviation of Gaussian noise (sigma) increases compared with the powerful lifting-based method MotionBERT (Zhu et al., 2023) while being more efficient.

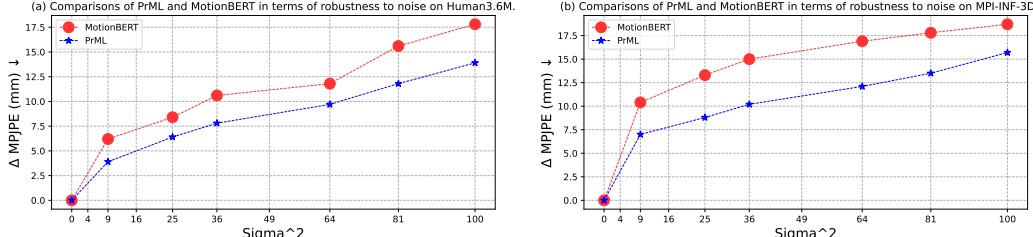

Figure 5: Comparisons of PrML and MotionBERT (Zhu et al., 2023) in terms of robustness to noise on Human3.6M and MPI-INF-3DHP datasets. Zero-mean Gaussian noise of standard deviation sigma is added to ground truth 2D pose, and we show their performance drop ( $\Delta$ MPJPE, in millimeters) as sigma increases. The size of markers indicates the computational cost of models.

### 5.3 ABLATION STUDY

We perform extensive ablation studies focused on analyzing the contribution of each component in our proposed PrML. Experiments are conducted on the Human3.6M (Ionescu et al., 2013) dataset with T = 243 as the number of input frames and MPJPE is used as the evaluation metric.

**Analysis on Effectiveness of Components.** The results in Section 5.2 have demonstrated that our framework achieves better results compared to the conventional lifting-based framework in terms of accuracy and robustness with fewer parameters. In this section, we show how to construct our proposed progressive multi-task learning pose estimation framework step by step.

- **Baseline:** Most multi-task learning models have a shared-bottom model structure following (Caruana, 1997) which substantially reduces the risk of overfitting (Ma et al., 2018). We follow this design and use the DSTFormer block from MotionBERT (Zhu et al., 2023) as the shared bottom. As shown in Table 4, this structure achieves 50.6mm MPJPE and serves as our baseline.
- **Multi-Task Branch:** To reduce the explicit influence of uncertain depth features on the well-detected 2D pose features, we introduce the multi-task branch design to refine the 2D pose features and learn the depth features separately. Based on the shared bottom, we incorporate the 2D pose branch and depth branch respectively bringing 6.2mm and 9.0mm error reduction (see also Table 4). With both 2D pose and depth branches, we achieve the MPJPE of 39.1mm.
- **Task-Aware Decoder:** As shown in Table 4, we achieve a reduction in MPJPE from 50.6mm to 46.2mm by adding task-aware decoder to the shared bottom. The performance is further improved to 38.6mm when task-aware decoder is introduced in conjunction with the multi-task branch.
- **PrML:** By introducing the mutual information loss to explicitly supervise the feature complement and the learning of bias, our TAD module further resulted in a 2.2mm error reduction (from 46.2mm to 44.0mm). After incorporating all components, we obtain the complete version of our PrML, which achieves the best performance with an MPJPE of 38.2mm.

Table 4: Analysis of the effectiveness of each component within PrML.

| Model Setting | Shared Bottom | 2D Pose Branch | Depth Branch | Task-Aware Decoder | Mutual Information | MPJPE ↓ |
|---|---|---|---|---|---|---|
| Baseline (Shared Bottom) | ✓ | - | - | - | - | 50.6 |
| + 2D Pose Branch Only | ✓ | ✓ | - | - | - | 44.4 (-6.2) |
| + Depth Branch Only | ✓ | - | ✓ | - | - | 41.6 (-9.0) |
| + TAD Only | ✓ | - | - | ✓ | - | 46.2 (-4.4) |
| + TAD + MI Loss | ✓ | - | - | ✓ | ✓ | 44.0 (-6.6) |
| + Multi-Task Branch | ✓ | ✓ | ✓ | - | - | 39.1 (-11.5) |
| + Multi-Task Branch + TAD | ✓ | ✓ | ✓ | ✓ | - | 38.6 (-12.0) |
| **PrML** | ✓ | ✓ | ✓ | ✓ | ✓ | 38.2 (-12.4) |

**Analysis on Task-Aware Decoder (TAD).** We first analyze the effectiveness of each operation in TAD and report performance in Table 5. By incorporating coarse alignment and task bias separately, we achieve the results of 39.5mm and 39.1mm. When both of them are incorporated, we obtain the best performance with an MPJPE of 38.2mm. We also examine the impact of different task biases within TAD. As shown in Table 6, the best results are achieved when both the 2D pose bias and per-joint bias are introduced and made learnable during the training process.

Table 5: Analysis of various designs within TAD.

| Step | Feature Complement | Coarse Alignment | Task Bias | MPJPE |
|---|---|---|---|---|
| 1 | ✓ | - | - | 39.5 |
| 2 | ✓ | ✓ | - | 39.1 |
| 3 | ✓ | - | ✓ | 38.7 |
| TAD | ✓ | ✓ | ✓ | 38.2 |

Table 6: Analysis of task biases within TAD.

| Step | 2D Pose Bias | Per-Joint Depth Bias | Learnable | MPJPE |
|---|---|---|---|---|
| 1 | ✓ | - | ✓ | 38.7 |
| 2 | - | ✓ | ✓ | 38.5 |
| 3 | ✓ | ✓ | - | 38.5 |
| TAD | ✓ | ✓ | ✓ | 38.2 |

**Multi-Task Learning for Lifting-Based Methods.** An alternative to our framework is to directly copy a branch from the lifting-based methods to construct the multi-task learning framework. However, this would increase the number of parameters substantially and lead to an unfair comparison. In light of this, we perform an embarrassingly simple transformation: replacing the original single regression head of the lifting-based framework with two regression heads. This replacement transforms the lifting-based framework into a hard parameter sharing multi-task learning framework (Ruder, 2017). As shown in Table 7, such simple modification leads to performance improvement in both multi-frames (MixSTE (Zhang et al., 2022b), MotionBERT (Zhu et al., 2023)) and single-frame (CA-PF (Zhao et al., 2023a)) lifting-based methods. Due to space limitations, we present more details in Appendix C.1.

Table 7: Analysis on the generalization of multi-task learning framework. (*) denotes our re-implementation.

| Method | Framework | MPJPE ↓ |
|---|---|---|
| MixSTE* (Zhang et al., 2022b) | Lifting-Based | 40.9 |
| + Two Regression Heads | Multi-Task Learning | 40.0 (-0.9) |
| MotionBERT* (Zhu et al., 2023) | Lifting-Based | 39.8 |
| + Two Regression Heads | Multi-Task Learning | 38.9 (-0.9) |
| CA-PF* (Zhao et al., 2023a) | Lifting-Based | 41.4 |
| + Two Regression Heads | Multi-Task Learning | 40.2 (-1.2) |

## 5.4 VISUALIZATION

**Feature Distribution Visualization.** To further analyze the difference between the 2D pose features and per-joint features, we utilize t-SNE (Van der Maaten & Hinton, 2008) to visualize their feature distributions. We select samples from Human3.6M (Ionescu et al., 2013) and visualize the features before regression heads (i.e., $\widetilde{F}_{2D}$ and $\widetilde{F}_D$). As shown in Figure 6, the distributions of the 2D features and depth features are different across various situations. These qualitative results provide strong evidence that we should not simply encode the 2D pose features and per-joint depth features in an entangled feature space. Due to space limitations, we present more visualization in our Appendix F.

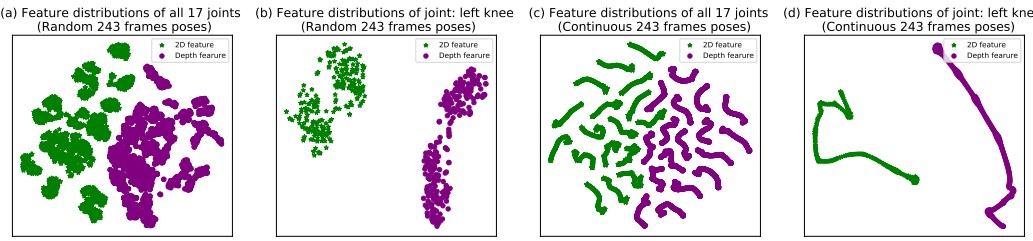

Figure 6: Feature distributions visualization of 2D features (green) and depth features (purple) using t-SNE (Van der Maaten & Hinton, 2008) method on Human3.6M (Ionescu et al., 2013) dataset. The distributions of the 2D features and depth features are different across various situations.

**3D Human Pose Estimation Visualization.** We present 3D human pose estimation results by our proposed PrML and MotionBERT (Zhu et al., 2023). As shown in Figure 7, our method generalizes well to in-the-wild videos including self-occlusion and fast motion.

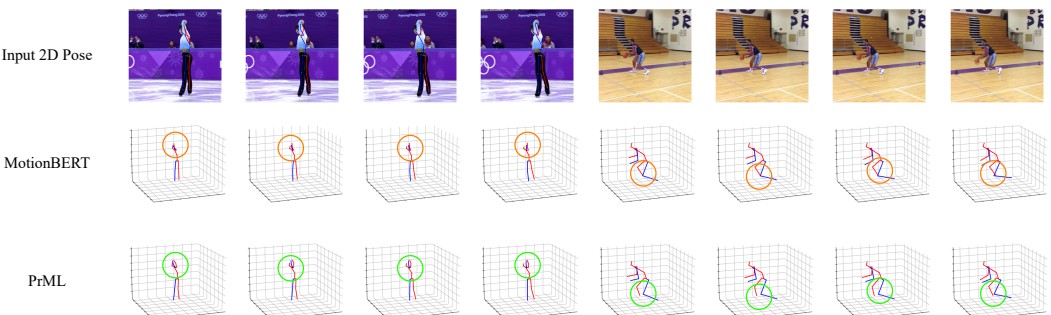

Figure 7: Qualitative comparisons of PrML with MotionBERT (Zhu et al., 2023). The green cycle indicates locations where our method achieves better results. See Appendix F for more comparison.

## 6 CONCLUSION

This work presents a novel progressive multi-task learning framework named PrML for monocular 3D human pose estimation. Our framework addresses the limitation of the lifting-based framework that neglects different initial states between a well-detected 2D pose and an unknown per-joint depth. PrML first learns 2D pose features and per-joint depth features separately by multi-task branch design and then employs a task-aware decoder to indirectly supplement information between the refined 2D pose features and the learned per-joint depth features. We also propose a mutual information loss to supervise the feature complementary process. Extensive quantitative experimental results on the Human3.6M and MPI-INF-3DHP datasets show that our PrML outperforms the conventional lifting-based framework in terms of accuracy and robustness with fewer parameters.

**Future Work.** The core contribution of our work is providing a new framework for monocular 3D human pose estimation. To this end, we use the widely used spatial and temporal transformer as our encoder to ensure a fair comparison with the lifting-based framework. It will be novel and interesting to design specific encoders for different tasks to extend our framework in future research.

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

# A APPENDIX

The Appendix is organized as follows:

# B EXPERIMENT SETTING

## B.1 DATASETS AND EVALUATION METRICS

**Human3.6M** (Ionescu et al., 2013) is the most popular benchmark for indoor 3D human pose estimation, which contains approximately 3.6 million frames captured by 4 cameras at different views. This dataset contains 11 subjects performing 15 typical actions (e.g., walking and sitting). To ensure a fair comparison, we follow previous methods (Zheng et al., 2021; Zhang et al., 2022b; Tang et al., 2023; Zhu et al., 2023) by using subjects 1, 5, 6, 7, and 8 for model training and subjects 9 and 11 for evaluation.

**MPI-INF-3DHP** (Mehta et al., 2017) is a recently proposed large-scale challenging dataset with both indoor and outdoor scenes. The training set comprises 8 subjects, covering 8 activities, ranging from walking and sitting to complex exercise poses and dynamic actions. The test set covers 7 activities, containing three scenes: green screen, non-green screen, and outdoor environments. It complements existing test sets with more diverse motions (standing/walking, sitting/reclining, exercise, sports (dynamic poses), on the floor, dancing/miscellaneous).

**Evaluation Metrics.** For the Human3.6M dataset, we use two common evaluation metrics: MPJPE and P-MPJPE. MPJPE (Mean Per Joint Position Error) is computed as the mean Euclidean distance between the estimated joints and the ground truth in millimeters after aligning their root joints (hip). P-MPJPE (Procrustes-MPJPE) is the MPJPE after the estimated joints align to the ground truth via a rigid transformation. For the MPI-INF-3DHP dataset, following previous works (Shan et al., 2022; Tang et al., 2023; Chen et al., 2023; Zhu et al., 2023), we use ground truth 2D pose as input and report MPJPE, Percentage of Correct Keypoint (PCK) with the threshold of 150mm, and Area Under Curve (AUC) as the evaluation metrics.

## B.2 IMPLEMENTATION DETAILS

Our model is implemented using PyTorch and executed on a server equipped with 2 NVIDIA 3090 GPUs. We apply horizontal flipping augmentation for both training and testing following (Tang et al., 2023; Zhu et al., 2023; Foo et al., 2023; Zhao et al., 2023a). For model training, we set each mini-batch as 16 sequences. The network parameters are optimized using AdamW (Loshchilov & Hutter, 2017) optimizer over 90 epochs with a weight decay of 0.01. The initial learning rate is set to 5e-4 with an exponential learning rate decay schedule and the decay factor is 0.99. In the experiments on Human3.6M, two kinds of input are utilized, including the 2D ground truth and the Stacked Hourglass (Newell et al., 2016) 2D pose detection, following (Zhu et al., 2023; Ci et al., 2019). For MPI-INF-3DHP, 2D ground truth is used following previous works (Zheng et al., 2021; Zhang et al., 2022b; Shan et al., 2022; Zhu et al., 2023). While our proposed framework is capable of adapting to input sequences of any length, to be fair, we choose specific input sequence lengths (denoted as $T$) for two datasets to compare our method with other approaches that have a certain 2D input length (Zheng et al., 2021; Zhang et al., 2022b; Shan et al., 2022; Tang et al., 2023): Human3.6M ($T = 81, 243$), MPI-INF-3DHP ($T = 9, 27, 81$).

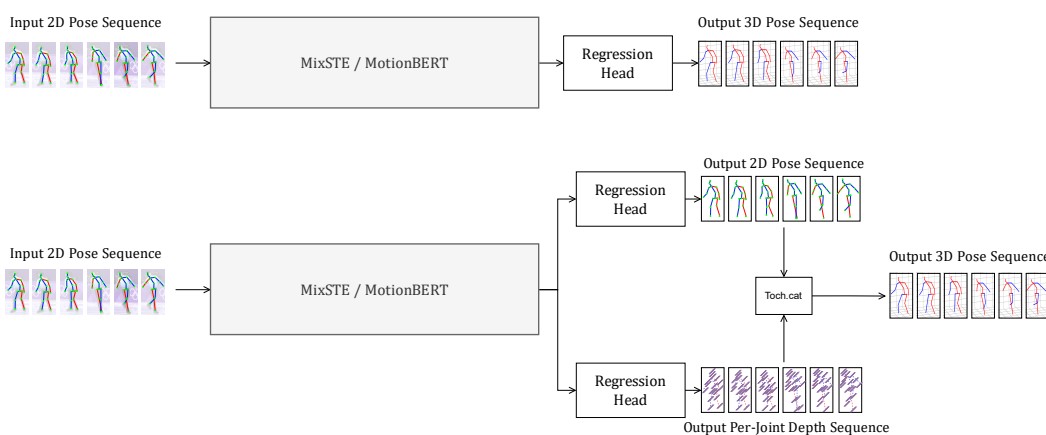

Figure 8: Previous pipeline (MixSTE (Zhang et al., 2022b) or MotionBERT (Zhu et al., 2023)): Using a single regression head to directly estimate the 3D pose. Improved pipeline: Replacing the original single regression head with two regression heads to transform the existing lifting framework into a hard parameter sharing multi-task learning framework (Ruder, 2017).

# C ADDITIONAL EXPERIMENT ANALYSIS

## C.1 MULTI-TASK LEARNING FOR LIFTING-BASED METHODS.

As shown in Figure 8, the 2D-to-3D lifting process employs a single regression head to directly estimate the 3D pose after extracting spatio-temporal information. We transform the existing 2D-to-3D lifting framework into a hard parameter sharing multi-task learning framework (Ruder, 2017) by replacing the original single regression head with two regression heads. We regress the

Table 8: Analysis on the generalization of multi-task learning framework. (*) denotes our re-implementation.

| Method | Framework | MPJPE |
|---|---|---|
| MixSTE* (Zhang et al., 2022b) | Lifting-Based | 40.9 |
| + Two Regression Heads | Multi-Task Learning | 40.0 ↓ 0.9 |
| MotionBERT* (Zhu et al., 2023) | Lifting-Based | 39.8 |
| + Two Regression Heads | Multi-Task Learning | 38.9 ↓ 0.9 |
| CA-PF* (Zhao et al., 2023a) | Lifting-Based | 41.4 |
| + Two Regression Heads | Multi-Task Learning | 40.2 ↓ 1.2 |

2D pose and per-joint depth separately. Then, we concatenate them together to obtain the 3D pose. As shown in Table 8, such simple modification leads to improvement in multi-frames (MixSTE (Zhang et al., 2022b), MotionBERT (Zhu et al., 2023)) and single-frame (CA-PF (Zhao et al., 2023a)) lifting-based methods.

Table 9: Analysis on the initial distribution of Task Bias within Task-Aware Decoder. G and L represent Gaussian distribution and Laplacian distribution respectively.

| Step | 2D Pose Bias | Per-Joint Depth Bias | MPJPE |
|---|---|---|---|
| 1 | L | G | 38.9 |
| 2 | G | L | 38.6 |
| 3 | L | L | 38.4 |
| Ours | G | G | 38.2 |

Table 10: Analysis on various micro designs within transformer block. $S$ and $T$ denote Spatial Transformer and Temporal Transformer.

| Step | S | T | T–S | S→T | T→S | MPJPE |
|---|---|---|---|---|---|---|
| 1 | ✓ | | | | | 40.6 |
| 2 | | ✓ | | | | 39.6 |
| 3 | | | ✓ | | | 39.0 |
| 4 | | | | ✓ | | 38.3 |
| Ours | | | | | ✓ | 38.2 |

## C.2 ANALYSIS ON TASK BIAS WITHIN TASK-AWARE DECODER.

In this section, we investigate the impact of different initial distributions of task bias on the model. As shown in Table 9, we use Gaussian or Laplace distribution as our initial distributions. We achieve a result of 38.9mm when initializing the 2D pose bias with Laplace distribution and the

per-joint depth with Gaussian distribution. The MPJPE decreased from 38.9mm to 38.6mm when we initialized the 2D pose bias with Gaussian distribution and the per-joint depth with Laplace distribution. Experimental results revealed that employing Gaussian distribution for both biases yielded the best performance.

### C.3 ANALYSIS ON STRUCTURE DESIGN OF TRANSFORMER BLOCK.

We conducted experiments on Human3.6M for five different types, including $S, T, S - T, S \to T$, $T \to S$ in the Transformer layer, where $S$ and $T$ denote Spatial Transformer Encoder and Temporal Transformer Encoder, respectively. $S - T$ is a two-stream layer that uses Spatial Transformer and Temporal Transformer Encoder simultaneously. $S \to T$ means Spatial Transformer first and then Temporal Transformer; the same goes for $T \to S$. The results in Table 10 show that the $T \to S$ variant has improved by 2.4mm (from 40.6mm to 38.2mm), 1.4mm (from 39.6mm to 38.2mm), 0.8mm (from 39.0mm to 38.2mm), and 0.1mm (from 38.3mm to 38.2mm), respectively, compared to the other four variants.

### C.4 ANALYSIS ON HYPERPARAMETER SETTING.

We consider the dimension $C$ of hidden feature, and the weight $\lambda_{MI}$ of mutual information loss as free parameters. Table 11 shows the results of different hyperparameter settings. We divide the configurations into 2 groups by row, and allocate different values for one hyperparameter in each group, while keeping the other hyperparameter fixed, to evaluate the impact of each configuration. The best results were obtained when the C was 128 and $\lambda_{MI}$ was 0.01. Based on the results shown in the table, we chose the combination of C = 128, and $\lambda_{MI}$ = 0.01 as our configuration.

Table 11: Analysis on the hyperparameter setting.

| Dimension (C) | $\lambda_{MI}$ | MPJPE |
|---|---|---|
| 64 | 0.01 | 40.4 |
| 96 | 0.01 | 39.3 |
| **128** | 0.01 | 38.2 |
| 128 | 1 | 38.6 |
| 128 | 0.1 | 38.5 |
| 128 | **0.01** | 38.2 |

### C.5 ADDITIONAL QUANTITATIVE RESULTS.

In this section, we provide the results of P-MPJPE on the Human3.6M (Ionescu et al., 2013) dataset. As shown in Table 12, our proposed PrML achieved the best performance with the MPJPE of 38.2mm and the third-best results with the P-MPJPE of 32.3mm.

Table 12: Results on Human3.6M in millimeters (mm) under MPJPE. T is the number of input frames. Seq2seq refers to estimating 3D pose sequences rather than only the center frame. MACs/frames represents multiply-accumulate operations for each output frame. The best result is shown in bold, and the second-best result is underlined.

| Method | Venue | Framework | Seq2Seq | T | Parameter | MACs | MACs/frame | MPJPE | P-MPJPE |
|---|---|---|---|---|---|---|---|---|---|
| MHFormer (Li et al., 2022b) | CVPR'22 | Lifting-Based | × | 351 | 30.9M | 7.1G | 7096M | 43.0 | 34.4 |
| MixSTE (Zhang et al., 2022b) | CVPR'22 | Lifting-Based | ✓ | 243 | 33.6M | 139.0G | 572M | 40.9 | 32.6 |
| P-STMO (Shan et al., 2022) | ECCV'22 | Lifting-Based | × | 243 | 6.2M | 0.7G | 740M | 42.8 | 34.4 |
| PoseFormerV2 (Zhao et al., 2023b) | CVPR'23 | Lifting-Based | × | 243 | 14.3M | 0.5G | 528M | 45.2 | 35.6 |
| STCFormer (Tang et al., 2023) | CVPR'23 | Lifting-Based | ✓ | 243 | 4.7M | 19.6G | 80M | 41.0 | 32.0 |
| GLA-GCN (Yu et al., 2023) | ICCV'23 | Lifting-Based | × | 243 | 1.3M | 1.5G | 1556M | 44.4 | 34.8 |
| MotionBERT (Zhu et al., 2023) | ICCV'23 | Lifting-Based | ✓ | 243 | 42.5M | 174.7G | 719M | 39.2 | - |
| KTPFormer (Peng et al., 2024) | CVPR'24 | Lifting-Based | ✓ | 243 | 33.7M | 69.5G | 286M | 40.1 | **31.9** |
| MotionAGFormer (Soroush Mehraban, 2024) | WACV'24 | Lifting-Based | ✓ | 243 | 19.0M | 78.3G | 322M | 38.4 | 32.5 |
| **PrML** (Ours) | - | Multi-Task Learning | ✓ | 243 | 13.0M | 49.3G | 203M | **38.2** | 32.3 |

### C.6 ATTEMPT TO USE IMAGE FEATURE.

As the inherent information entropy of the input 2D pose sequence, the performance of methods only using 2D pose sequences as input gradually approaches the theoretical limit. Incorporating image features is a potential solution to this general limitation. There are several single-frame lifting-based methods that utilize image features, such as ContextPose (Ma et al., 2021a)and CA-PF (Zhao et al., 2023a). Unfortunately, integrating these methods into existing multi-frame lifting methods or our PrML is challenging due to their complex model architectures. Therefore, we employ one cross-attention layer to fuse image features, making a preliminary attempt to fuse image features into our

framework. Due to the lightweight nature of 2D poses, traditional lifting-based frameworks typically input a 2D pose sequence. However, inputting a long-term image sequence is clearly impractical. How to effectively extend image input from a single frame to multiple frames remains an open question. To tackle this problem, we employ VAE to compress the original images into latent features, allowing for multi-frame input. We use the pre-trained VAE (Kingma, 2013) from SDXL (Podell et al., 2023). Although we have compressed the images using VAE, processing 243 frames remains computationally expensive. Therefore, we leverage VAE to incorporate 15 frames image feature, making a preliminary attempt to fuse multi-frame image features into our framework.

As shown in the Table 13, despite using simple cross-attention to fuse image features, we still observe performance improvement. When we incorporated image features into the 2D branch, we observed a significant improvement. This is likely due to the fact that image features can easily capture 2D spatial information. However, when we added image features to the depth branch, the improvement was limited. This is because we used a simple cross-attention mechanism to process the images, and extracting depth information from RGB images is a challenging task. Similarly, when we incorporated image features into all branches, the improvement was modest. These experiments demonstrate the potential for further extending our framework.

Moreover, the insights from multi-task learning can be further extended. Using 2D pose detectors to estimate 2D poses has been widely used, so why not similarly utilize powerful depth estimation networks (like DepthAnything (Yang et al., 2024)) to estimate a relative depth from the image to facilitate 3D human pose estimation as well? We can integrate depth features into existing single-frame methods that utilize image features to validate this idea. As shown in Table 14, by incorporating depth features from DepthAnything (Yang et al., 2024), we can reduce the error of CA-PF by 1.9mm, which demonstrates the effectiveness of depth features.

In summary, how to leverage image features to facilitate multi-frame 3D human pose estimation is a non-trivial question and our future research direction.

Table 13: Results on the Human3.6M in millimeters (mm) under MPJPE. T is the number of input frames.

| Method | T | MPJPE |
|---|---|---|
| PrML | 15 | 49.7 |
| + Image Feature for 2D | 15 | 47.2 |
| + Image Feature for Depth | 15 | 49.6 |
| + Image Feature for All | 15 | 49.5 |

Table 14: Results on the Human3.6M in millimeters (mm) under MPJPE.

| Method | MPJPE |
|---|---|
| CA-PF (Zhao et al., 2023a) | 41.4 |
| + Depth Feature (Yang et al., 2024) | 39.5 |

Table 15: Results on the 3DPW in millimeters (mm) under MPJPE and P-MPJPE.

| Method | MPJPE | P-MPJPE |
|---|---|---|
| MotionBERT (Zhu et al., 2023) | 85.5 | 50.2 |
| MotionBERT (Zhu et al., 2023) | 76.9 | 47.2 |
| PrML | 73.9 | 49.1 |

### C.7 ADDITIONAL EXPERIMENT ON 3DPW.

We also conduct an experiment on the 3DPW dataset following the experiment setting of Motion-BERT (Zhu et al., 2023). As other methods we compared did not conduct experiments on 3DPW, we are unable to provide their performance in the table. As shown in the Table 15, the MPJPE of our PrML is not only lower than the MotionBERT trained from scratch but also lower than its finetune variant. PrML's P-MPJPE is still lower than the MotionBERT trained from scratch.

## D LIMITATION.

Although we propose a novel framework for monocular 3D human pose estimation, our input remains the same as conventional lifting-based methods: a 2D pose sequence. Such 2D joint coordinates undoubtedly cause visual representation reduction compared to raw images. This inherent information entropy of the 2d pose sequence limits the theoretical upper bound on performance.

## E ADDITIONAL ANALYSIS OF MUTUAL INFORMATION

Calculating the conditional mutual information loss is notoriously difficult, especially in neural networks (Hjelm et al., 2018; Tian et al., 2021). The approximate mutual information loss is much easier to compute. Given $A \geq B$, maximizing B can maximize the lower bound of A. By maximizing the approximate mutual information, we can maximize the lower bound of the conditional mutual

information. This is in line with our original goal: provide as much relevant information as possible. In summary, although it is a mathematically suboptimal result, it is the best result we can achieve.

We use $Y$ to represent the label, $S$ to represent task support features and $B$ to represent task bias. Optimizing this objective will maximize the mutual information between task support features and the label to support the task. By the definition of condition mutual information, we have:

$$
\begin{aligned}
\mathcal{I}(Y; S \mid B) &= \int_B \left( \int_S \int_Y p_{Y,S|B}(y,s|b) \log \left( \frac{p_{Y,S|B}(y,s|b)}{p_{Y|B}(y|b)p_{S|B}(s|b)} \right) dyds \right) p_B(b)db \\
&= \int_S \int_Y p(y,s) \log \left( \frac{p(y,s)}{p(y)p(s)} \right) dyds - \int_B \int_S p(s,b) \log \left( \frac{p(s,b)}{p(s)p(b)} \right) dsdb \\
&\quad + \int_Y \left( \int_B \int_S p_{S,B|Y}(s,b|y) \log \left( \frac{p_{S,B|Y}(s,b|y)}{p_{S|Y}(s|y)p_{B|Y}(b|y)} \right) dsdb \right) p_Y(y)dy \quad (10) \\
&= \mathcal{I}(Y; S) - \mathcal{I}(S; B) + \int_Y \underbrace{D_{KL}(P_{(S,B)|Y} \| P_{S|Y} P_{B|Y})}_{\text{KL Divergence} \geq 0} dP_Y \\
&= \mathcal{I}(Y; S) - \mathcal{I}(S; B) + \mathbb{E}_Y[D_{KL}(P_{(S,B)|Y} \| P_{S|Y} P_{B|Y})] \\
&\geq \mathcal{I}(Y; S) - \mathcal{I}(S; B)
\end{aligned}
$$

The optimization objective thus becomes:

$$
\max \mathcal{I}(Y; S \mid B) \longrightarrow \max \mathcal{I}(Y; S) - \mathcal{I}(S; B) \quad (11)
$$

However, since both $\mathcal{I}(Y; S)$ and $\mathcal{I}(S; B)$ are non-negarive (KL Divergence $\geq 0$), the $\mathcal{I}(Y; S) - \mathcal{I}(S; B)$ will result in negative values during training. This will cause the training process to fail to converge. Therefore, we simplified the implementation of mutual information by calculating only the first term. Our final optimization objective becomes:

$$
\max \mathcal{I}(Y; S \mid B) \longrightarrow \max \mathcal{I}(Y; S) \quad (12)
$$

## F  ADDITIONAL VISUALIZATION

To further comparing the accuracy of the projected 2D pose, we report the PCKh@0.5 following MPII (Andriluka et al., 2014). The table shows that our PrML achieves better accuracy in the projected 2D poses than the MotionBERT.

Table 16: Results on Human3.6M. (PCKh@0.5)

| Method | PCKh@0.5 |
|---|---|
| MotionBERT | 95.9 |
| PrML | 96.4 |

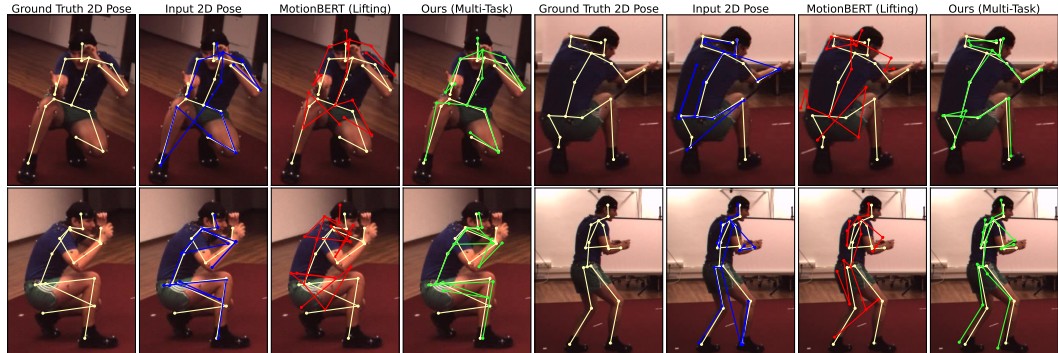

Figure 9: Qualitative Comparison of 2D Pose (Ground Truth, Input, MotionBERT (Zhu et al., 2023) and Ours). We project the 2D pose in the camera coordinate system back to the image coordinate system for comparison. The powerful lifting-based method MotionBERT gets a 2D pose worse than the input, which contradicts our intuition. In contrast, our proposed framework obtains a 2D pose better than the input.

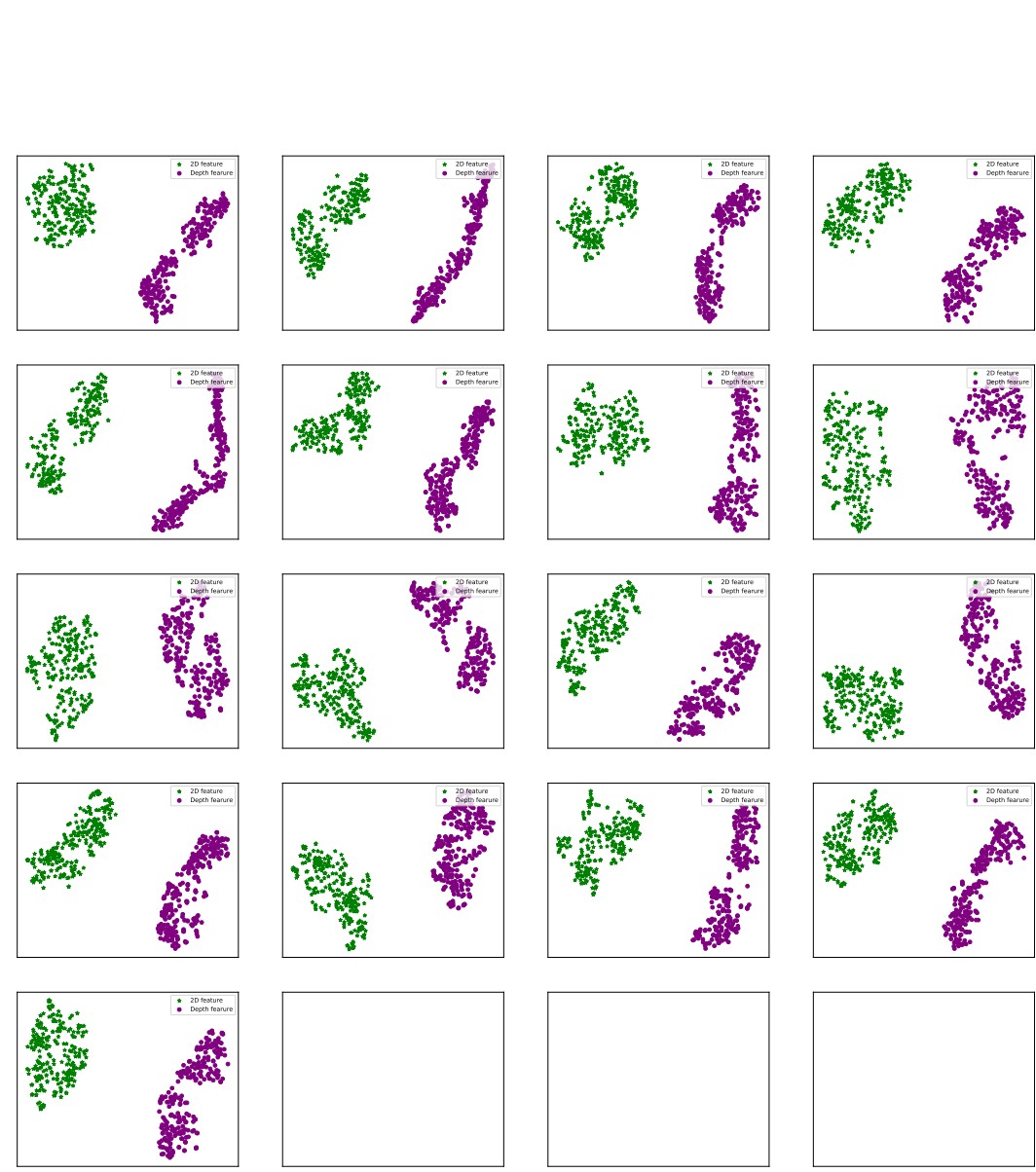

Figure 10: Feature distributions visualization of 2D feature (green) and depth feature (purple) using t-SNE (Van der Maaten & Hinton, 2008) method on Human3.6M (Ionescu et al., 2013) dataset. (Random 243 frames poses, all 17 joints are visualized)

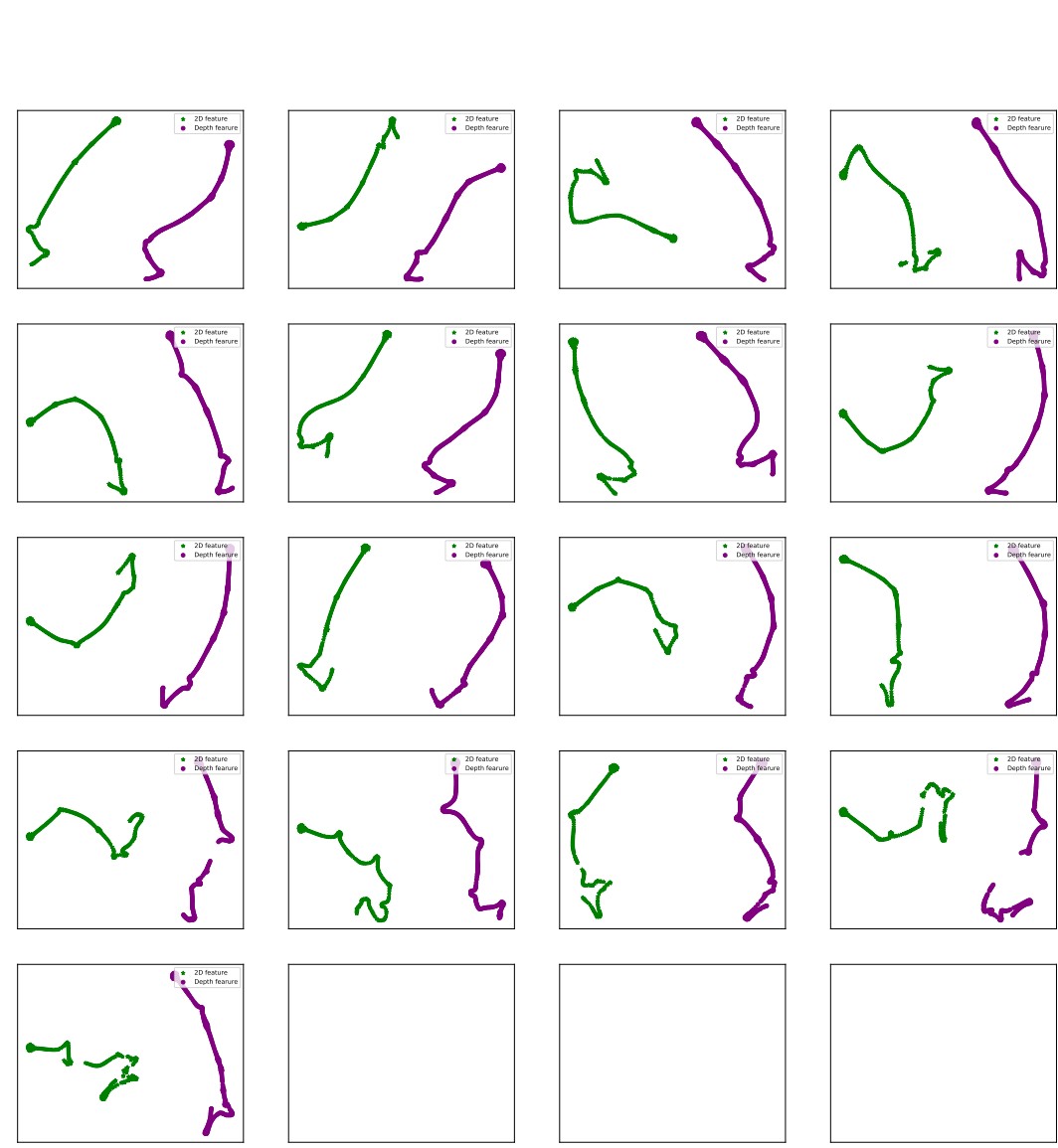

Figure 11: Feature distributions visualization of 2D feature (green) and depth feature (purple) using t-SNE (Van der Maaten & Hinton, 2008) method on Human3.6M (Ionescu et al., 2013) dataset. (Continuous 243 frames poses, all 17 joints are visualized)

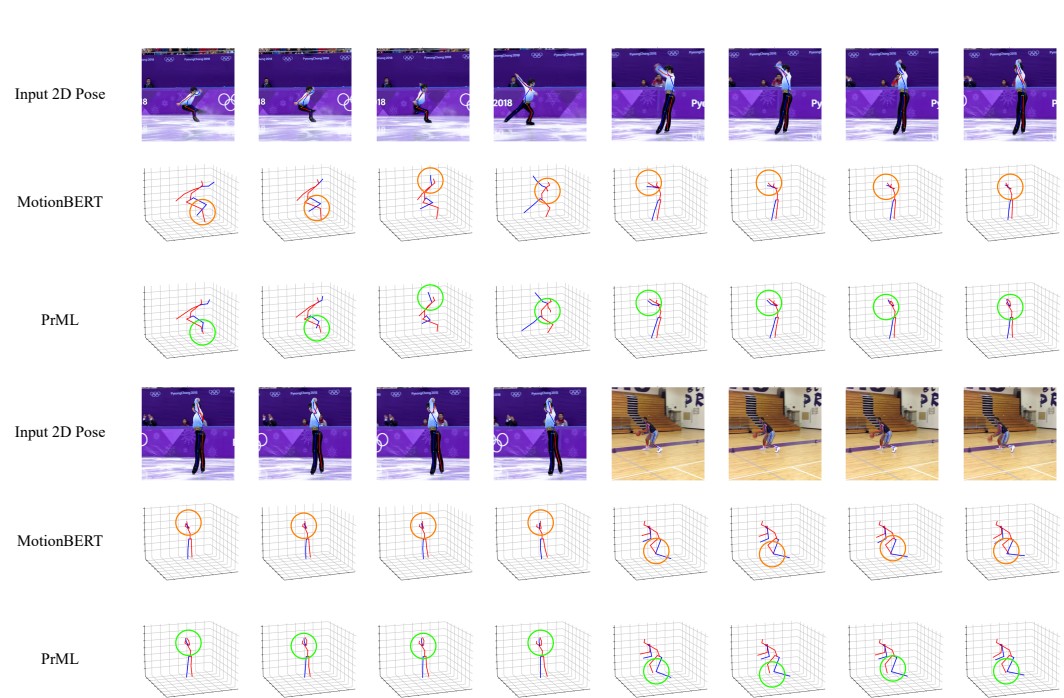

Figure 12: Qualitative comparisons of PrML with MotionBERT (Zhu et al., 2023). The green cycle indicates locations where our method achieves better results.

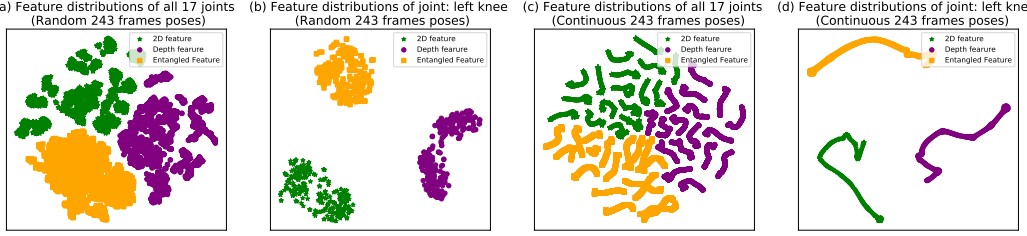

Figure 13: Feature distributions visualization of 2D features (green) and depth features (purple) and entangled features (orange) using t-SNE (Van der Maaten & Hinton, 2008) method on Human3.6M (Ionescu et al., 2013) dataset. The 2D features and depth features are extracted from our PrML (Multi-Task Learning). The entangled features are extracted from the lifting-based method MotionBERT (Zhu et al., 2023). Visualization results show that the distribution of entangled features is more compact, while our 2D pose features and per-joint depth features exhibit a more diverse distribution, indicating the potential to learn various representations.

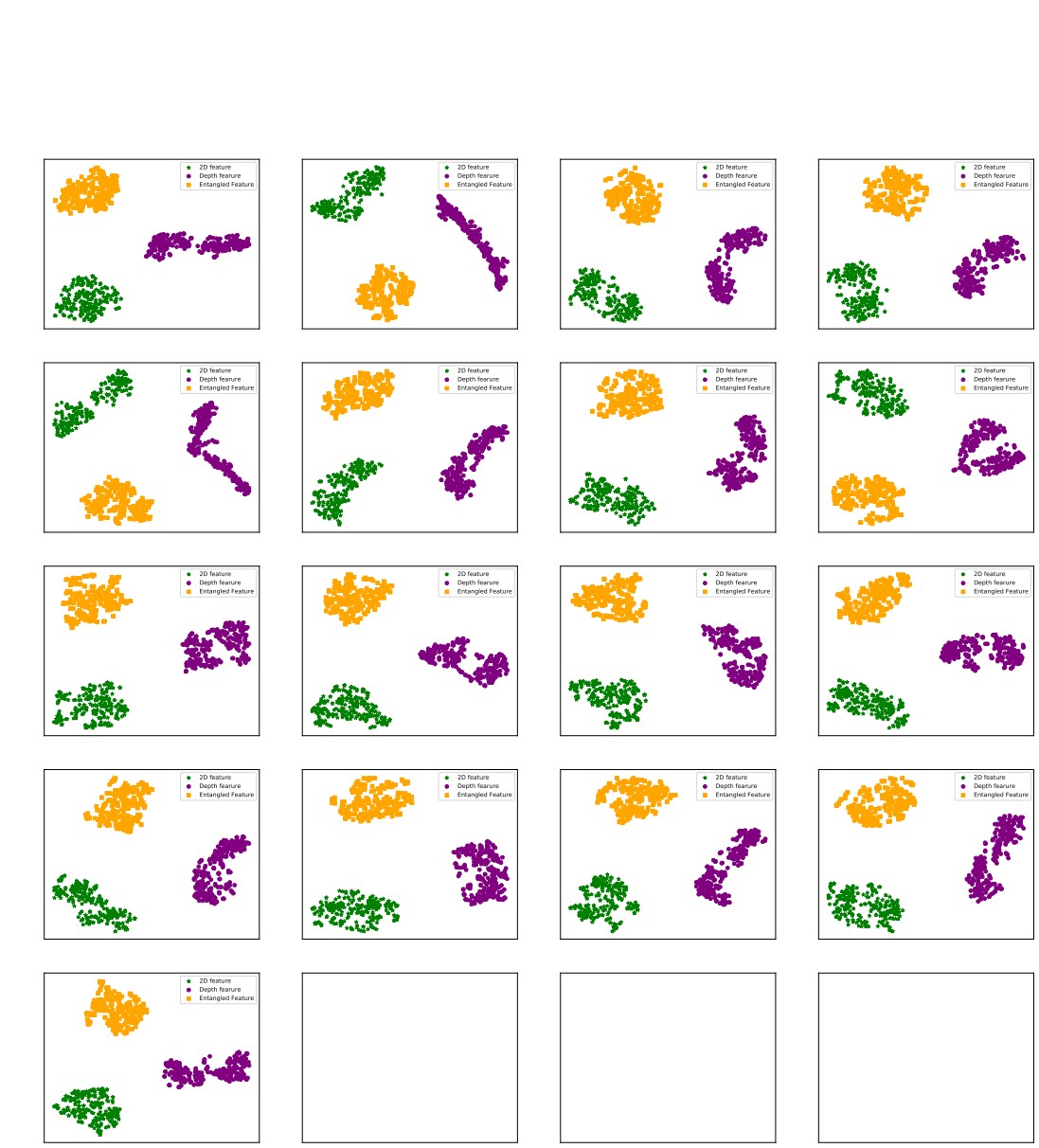

Figure 14: Feature distributions visualization of 2D feature, depth feature and entangled feature using t-SNE (Van der Maaten & Hinton, 2008) method on Human3.6M (Ionescu et al., 2013) dataset. (Random 243 frames poses, all 17 joints are visualized)

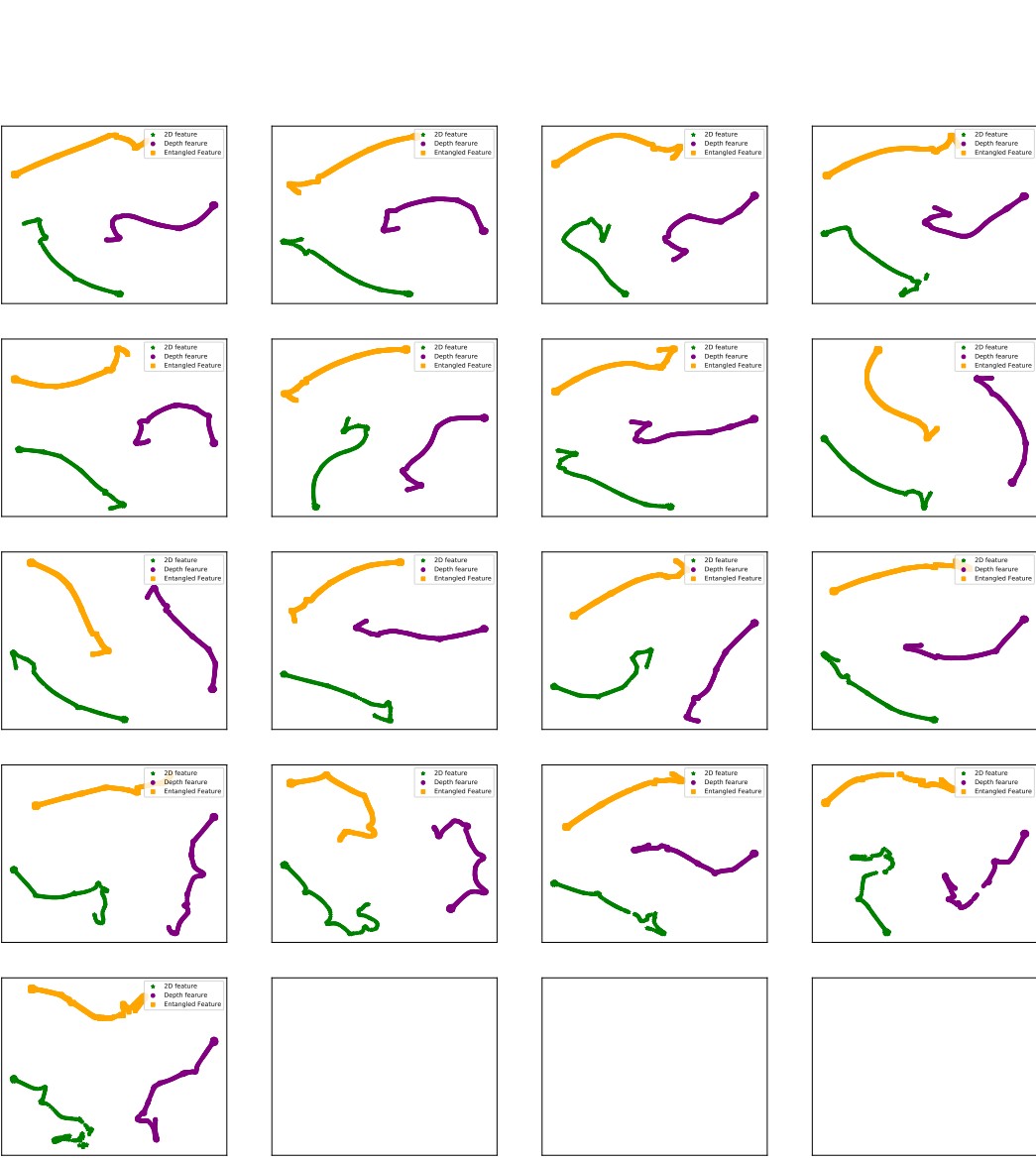

Figure 15: Feature distributions visualization of 2D feature, depth feature and entangled feature using t-SNE (Van der Maaten & Hinton, 2008) method on Human3.6M (Ionescu et al., 2013) dataset. (Continuous 243 frames poses, all 17 joints are visualized)

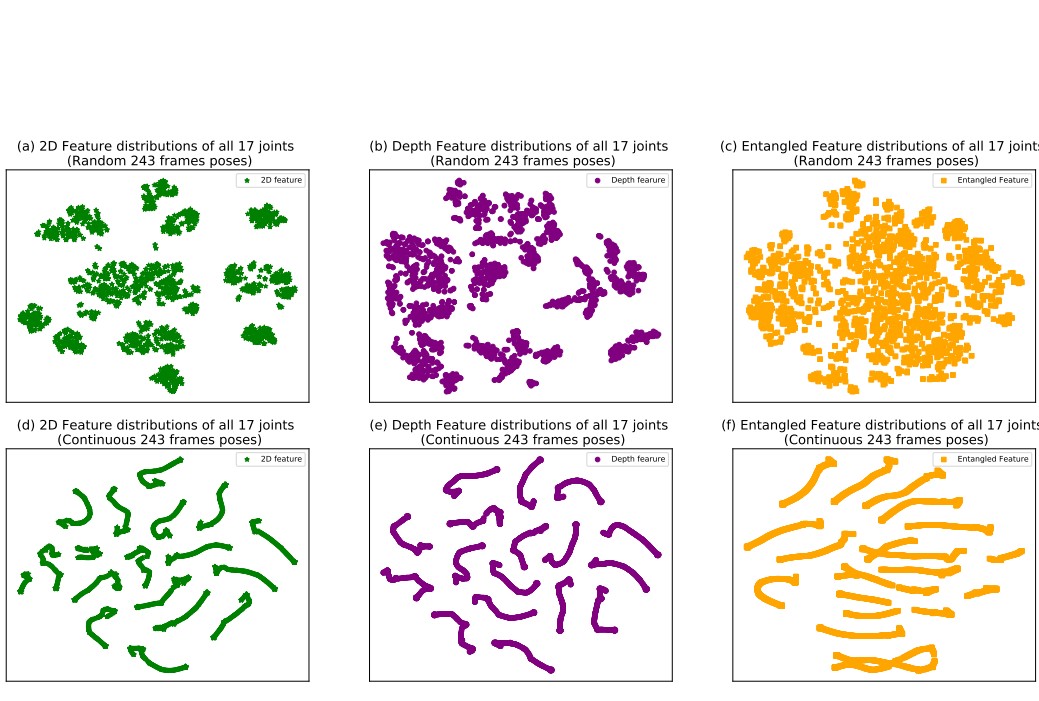

Figure 16: Feature distributions visualization of 2D feature, depth feature, and entangled feature in their own feature space using t-SNE (Van der Maaten & Hinton, 2008) method on Human3.6M (Ionescu et al., 2013) dataset. (Random 243 frames poses, all 17 joints are visualized)

