# OpenReview forum: "PrML: Progressive Multi-Task Learning for Monocular 3D Human Pose Estimation"
_ICLR.cc/2025/Conference — Submitted to ICLR 2025_

### Official Review · Reviewer_zMrU · 2024-10-19

**Soundness:** 2
**Presentation:** 2
**Contribution:** 2
**Rating:** 3
**Confidence:** 5

**Summary:**

This paper presents a 3D human pose estimation system that takes a 2D pose as an input. Different from previous works that output 3D pose using a single module, the proposed one estimates refined 2D pose and depth using respective modules. The motivation for this is to preserve well-detected input 2D pose as much as possible and prevent the depth estimation module from making the encoded 2D pose bad. In addition, the task-aware decoder is introduced to minimize the gap between 2D and depth modules.

**Strengths:**

An attempt to preserve well-detected 2D poses makes sense.

**Weaknesses:**

1. Novelty is not enough. The main idea of this work is to preserve well-detected 2D pose when doing the 3D pose estimation. First, the idea of separately estimating 2D pose and depth have been tried many times. For example, “Hand Pose Estimation via Latent 2.5D Heatmap Regression” (ECCV 2018). Second, how the authors achieved this is too straightforward by simply designing separate branches. The architecture of each branch simply follows existing Transformer-based ones. The task-aware decoder is introduced to complement relevant information; however, Table 4 shows that introducing TAD improves the 3D errors less than 1 mm.

2. Not reliable ablation studies. Table 4 shows that the baseline has MPJPE 50, which is worse than any methods in Table 1. It seems the baseline is too weak and this exaggerates the effectiveness of the proposed contributions, such as multi-task learning. Table 7 shows that the improvement is more or less 1 mm. I’m not sure the 1 mm improvement is meaningful.

3. Old benchmarks. The authors use Human3.6M and MPI-INF-3DHP benchmarks. They are good, but old. They were published more than 10 years ago. The authors should have reported results on newer ones as well, including 3DPW, AGORA, BEDLAM, EMDB.

**Questions:**

No questions.

---

> ### Author Response · Authors · 2024-11-23
> **Authors Response to Reviewer #zMrU (Part 1/2)**
>
> Dear Reviwer #zMrU,
>
> Thanks for your valuable time in reviewing our work. We will address your concerns in the following part. The corresponding revised content is highlighted in red font within our updated paper.
>
> ---
>
> >**Weakness 1.1**: Novelty is not enough. The main idea of this work is to preserve well-detected 2D pose when doing the 3D pose estimation. First, the idea of separately estimating 2D pose and depth have been tried many times. For example, “Hand Pose Estimation via Latent 2.5D Heatmap Regression” (ECCV 2018).
>
> Thank you for sharing related work that separately estimates 2D pose and depth [1], we discuss this paper in our Related Work section in our updated manuscript. However, our method can be distinctively differentiated from it in the following aspects:
>
> **1>Motivation:** To address the **scale and depth ambiguity** in 3D hand pose estimation from a single image, [1] proposes a scale and translation invariant 2.5D pose representation. In contrast, our proposed progressive multi-task learning framework is designed to **mitigate the erosion of well-detected 2D pose caused by the uncertain per-joint depth**.
>
> **2>Different Research Task and Pipeline:** [1] reconstructs **3D hand pose from a single RGB image.** In contrast, our work focuses on monocular 3D human pose estimation. We first detect a 2D human pose sequence from a video using a human pose detector and then leverage this **2D pose sequence as input to estimate the 3D human pose sequence.**
>
> **3>Contribution:** The key contribution of [1] is a **novel 2.5D representation**, which faciliate 3D hand pose estimation from a monocular image. [1] also propose a **novel CNN architecture** to implicitly learn depth maps and heatmap distributions. In contrast, with the context of the lifting-based framework dominating the field of monocular 3D human pose estimation, we propose a **new progressive multi-task learning framework**, different from the previous work of designing encoders within the lifting-based framework.
>
> [1] Hand Pose Estimation via Latent 2.5D Heatmap Regression ECCV'18
>
> ---
>
> >**Weakness 1.2**: Second, how the authors achieved this is too straightforward by simply designing separate branches. The architecture of each branch simply follows existing Transformer-based ones.
>
> Thanks for your feedback. As we mentioned in our future work, **we use the widely used spatial and temporal transformer as our encoder to ensure a fair comparison with the lifting-based framework, which demonstrates that the improvements we achieved are attributed to our framework-level modifications, rather than encoder design.**
>
>
> The key contribution of our work primarily focuses on stepping outside of the conventional lifting-based framework to investigate the impact of the lifting process itself on monocular 3D human pose estimation, rather than the design of the encoder. Experimental results on the Human3.6M and MPI-INF-3DHP datasets (Table 1,2,3) show that our proposed progressive multi-task learning framework outperforms the conventional lifting-based framework in terms of accuracy and robustness with fewer parameters. It will be novel and interesting to design specific encoders for different tasks to extend our framework in future research.
>
> ---
>
> >**Weakness 1.3**: The task-aware decoder is introduced to complement relevant information; however, Table 4 shows that introducing TAD improves the 3D errors less than 1 mm.
>
> Table 4 presents two results to demonstrate the effectiveness of our proposed TAD. The configuration of "Shared Bottom + TAD" shows that our TAD brings a 4.4mm error reduction. The improvement was relatively modest with a model configuration of "Shared Bottom + Multi-Task Branch + TAD" due to the performance approaching its theoretical limit.
>
> ---
>
> >**Weakness 2.1**: Not reliable ablation studies. Table 4 shows that the baseline has MPJPE 50, which is worse than any methods in Table 1. It seems the baseline is too weak and this exaggerates the effectiveness of the proposed contributions, such as multi-task learning.
>
> Rather than modifying the encoder in existing lifting-based methods, we present a progressive multi-task learning framework that offers a new paradigm for this task. We start ablation studies with a simple shared bottom as a baseline to construct our multi-task learning framework step by step.
>
> The performance of the shared bottom is lower than the previous state-of-the-art method can be attributed to the relatively small feature dimension of 128. The shared bottom is used to learn a basic feature representation. In contrast, previous lifting methods in Table 1 typically set the feature dimension to 512.
>
>
> ---

---

> ### Author Response · Authors · 2024-11-23
> **Authors Response to Reviewer #zMrU (Part 2/2)**
>
> >**Weakness 2.2**: Table 7 shows that the improvement is more or less 1 mm. I’m not sure the 1 mm improvement is meaningful.
>
> Thanks for your feedback. Table 7 provides a **alternative** of how our multi-task learning concept can be applied to lifting methods. Conventional lifting-based methods directly regress the 3D pose using one regression head at the end of the model. In Table 7, we performed a very **simple transformation: replacing the original single regression head in the lifting-based framework with two regression heads**, one for 2D pose regression and the other for per-joint depth regression. Such simple replacement leads to performance improvement with little increase in the number of parameters for both multi-frames and single-frame methods.
>
> Additionally, the 1mm error reduction is a worthwhile improvement for monocular 3D human pose estimation. As shown in the table, the original MixSTE[3] achieved an MPJPE of 40.9mm. By simply replacing its single regression head with two regression heads, we achieved an MPJPE of 40.0mm, outperforming the latest method KTPFormer[8].
>
> |Method|Venue|MPJPE|
> |--|--|--|
> |MixSTE [3]|CVPR'22|40.9|
> |STCFormer [6]|CVPR'23|41.0|
> |KTPFormer [8]|CVPR'24|40.1|
> |MixSTE [3] + Two Regression Head||40.0|
>
> ---
>
> >**Weakness 3**: Old benchmarks. The authors use Human3.6M and MPI-INF-3DHP benchmarks. They are good, but old. They were published more than 10 years ago. The authors should have reported results on newer ones as well, including 3DPW, AGORA, BEDLAM, EMDB.
>
> According to your constructive advice, we conduct an experiment on the 3DPW dataset following the experiment setting of MotionBERT [1]. As other methods we compared (including MHFormer[2], MixSTE[3], P-STMO[4], PoseFormerV2[5], STCFormer[6], GLA-GCN[7], KTPFormer[8]) did not conduct experiments on 3DPW, we are unable to provide their performance in the table.
>
> |Method|MPJPE|P-MPJPE|
> |--|--|--|
> |MotionBERT[1] (scratch)|85.5|50.2|
> |MotionBERT[1] (finetune)|76.9|47.2|
> |PrML (Ours)|73.9|49.1|
>
>
> As shown in the table, the MPJPE of our PrML is not only lower than the MotionBERT trained from scratch but also lower than its finetune variant. PrML's P-MPJPE is still lower than the MotionBERT trained from scratch. We added this quantitative comparison in our Appendix.C.7 "Additional Experiment on 3DPW".
>
>
> [1] MotionBERT: A unified perspective on learning human motion representations. ICCV'23
>
> [2] MHFormer: Multi-hypothesis transformer for 3D human pose estimation. CVPR'22
>
> [3] MixSTE: Seq2seq Mixed Spatio-Temporal Encoder for 3D Human Pose Estimation in Video. CVPR'22
>
> [4] P-STMO: Pre-Trained Spatial Temporal Many-to-One Model for 3D Human Pose Estimation. ECCV'22
>
> [5] PoseFormerV2: Exploring frequency domain for efficient and robust 3D human pose estimation. CVPR'23
>
> [6] 3D Human Pose Estimation with Spatio-Temporal Criss-cross Attention. CVPR'23
>
> [7] GLA-GCN: Global-local Adaptive Graph Convolutional Network for 3D Human Pose Estimation from Monocular Video. ICCV'23
>
> [8] KTPFormer: Kinematics and Trajectory Prior Knowledge-Enhanced Transformer for 3D Human Pose Estimation. CVPR'24
>
> ---
>
> Thanks again for your valuable time and considerate advice, which help us improve the quality of our work. We hope our effort could address your concerns. We are more than willing to provide further clarifications if there are any lingering questions or concerns.
>
> Best!
>
> The authors of Paper #2366

---

> > ### Author Response · Authors · 2024-11-25
> > **Please let us know whether we have addressed all the issues**
> >
> > Dear reviewer,
> >
> > Thanks for the comments. We have provided more explanations and results to your questions. Please let us know whether we have addressed all the issues as the deadline (Nov 26) is fast approaching.
> >
> > Please also consider raising the score after all the issues are addressed.
> >
> > Thank you,

---

> > > ### Comment · Reviewer_zMrU · 2024-11-25
> > >
> > > 1. Weakness 1.1
> > >
> > > The suggested 2.5D representation is not only for addressing the depth and scale ambiguity. 2.5D means (x,y) in pixel space and depth in root-relative depth space. By calculating a 3D translation vector from a camera to the root joint, full 3D keypoint coordinates can be recovered from the 2.5D representation. By using their 2.5D representation, many methods, including 3DCrowdNet (CVPR 2022) successfully preserved well-detected input 2D pose (they even made the input 2D pose better by refining it with image features). The authors did not provide comparisons to any of such previous efforts to utilize input 2D pose better. The proposed baseline is too weak and experimental demonstration is quite limited (mostly done on old benchmarks and those benchmarks are not from in-the-wild environments).
> > >
> > > 2. Weakness 1.2
> > >
> > > Even the framework-level modifications are not novel. The motivation (preserving well-detected 2D pose) could be interesting, but the proposed system is not complete and showed very limited experimental demonstrations. As stated, the benchmarks are introduced more than 10 years ago, and all of them are not from in-the-wild environments. There should be much more extensive and interesting experimental demonstration to prove the contribution especially when the proposed idea is simple like this paper.
> > >
> > > 3. Weakness 1.3
> > >
> > > Improvements from baseline (50.6) to TAD only (46.2) is also not very meaningful as 50.6 error is higher than any methods in Table 1 and 2. That means, the baseline is too weak and exaggerated.
> > >
> > > 4. Weaknesses 2.1
> > >
> > > The rebuttal did not resolve this concern. A baseline, weaker than any methods in the comparison table, is definitely too weak and could exaggerate the proposed contributions.
> > >
> > > 5. Weaknesses 2.2
> > >
> > > I know that Table 7 shows that the proposed item could be used in other methods as well. But still, 1 mm improvements are marginal.
> > >
> > > 6. Weaknesses 3
> > >
> > > 3DPW is one of the most widely used in-the-wild benchmarks. Lots of works have been reporting their performance on 3DPW, including SPIN (ICCV 2019), I2L-MeshNet (ECCV 2020), Pose2Mesh (ECCV 2020), HybrIK (CVPR 2021), METRO (CVPR 2021), 3DCrowdNet (CVPR 2022), PyMaF (ICCV 2021), Humans in 4D (ICCV 2023). The authors should compare the proposed one with more diverse and unlimited way. Overall, experimental comparison and demonstrations are quite restricted to small cases.

---

### Official Review · Reviewer_AWXr · 2024-10-30

**Soundness:** 3
**Presentation:** 3
**Contribution:** 3
**Rating:** 6
**Confidence:** 4

**Summary:**

This paper points out the problem of current lifting-based 3D human pose estimation: encoding the well-detected 2D pose and uncertain depth will erode the 2D pose. Based on such intuition and some experimental observations, the authors propose to predict 2D pose and per-joint depth using two disentangled branches as a Multi-task learning task to solve the raised problem.
The authors adopt a task-aware decoder to share the 2D pose and depth features. Two types of support features are achieved by combining two corresponding learnable 2d pose and depth biases. The supervising and unsupervised mutual information losses are leveraged to train the model.
The extensive experiments show the proposed method gains improvements over SOTA lifting-based methods. And the ablations validate the effectiveness of each component. Feature distributions and pose visualization show significant results.

**Strengths:**

1. The paper is well-written and easy to follow.
2. The paper proposes a novel perspective for solving the 2d-3d lifting problem of 3d human pose estimation. It significantly focuses on the uncertainty problem of per-joint depth estimation. Predicting 2D pose and per-joint depth in a disentangled manner reduces the difficulty of prediction and may have a less negative impact on the regression of 2D pose coordinates.
3. The experiments and demonstrations are thorough and persuasive, and the effectiveness of each proposed module has been validated.
4. The experiments involving noisy 2D poses demonstrate the model's improved robustness, which aligns with the motivations behind the model design.

**Weaknesses:**

1. Although the modeling of 2D-3D lifting is significantly improved in this paper, the per-joint depth information implicitly contained in the 2D pose sequence remains limited.  As performance approaches the theoretical limit, the accuracy upper bound of depth prediction may become less related to the model architecture, due to the inherent information entropy of the input 2D pose sequence.
2. The accuracy gain brought by the mutual information loss is limited, as shown in Table 4.
3. The feature distribution visualization in Figure 6 reveals the disentangled states of pose and depth features within their feature space. However, since the 2D XY coordinates and the Z coordinate of the 3D pose have different value distributions, it is natural to see their original features distributed across each other. Some comparisons with other lifting-based methods may be needed.

**Questions:**

1. Did the authors attempt to use image features as input to improve the depth prediction capability, given that the lifting problem is still ill-posed?
2. What is the meaning of the statement regarding the different initial states of the 2D pose and per-joint depth? Does this indicate an error in the detected 2D pose?
3. The qualitative examples of the inaccurate projected 2D poses in Figure 2 may be insufficient. Some quantitative metrics comparing the accuracy of the projected 2D poses versus existing methods could provide stronger evidence.

---

> ### Author Response · Authors · 2024-11-23
> **Authors Response to Reviewer #AWXr (Part 1/3)**
>
> Dear Reviwer #AWXr,
>
> Thanks for your valuable time in reviewing our work. We will address your concerns in the following part. The corresponding revised content is highlighted in red font within our updated paper.
>
> ---
>
> >**Weakness 1**: Although the modeling of 2D-3D lifting is significantly improved in this paper, the per-joint depth information implicitly contained in the 2D pose sequence remains limited. As performance approaches the theoretical limit, **the accuracy upper bound of depth prediction may become less related to the model architecture, due to the inherent information entropy of the input 2D pose sequence.**
>
> We agree that the inherent information entropy of the input 2D pose sequence limits the theoretical upper bound on performance. This limitation is also prevalent in conventional lifting-based methods as their input is also a 2D pose sequence. **We have add a new section Appendix.D "Limitation" in our updated paper. (page 19)** The revised content is indicated in red. Please check our updated manuscript. Alternatively, you can directly examine the following content.
>
> "Limitation. Although we propose a novel framework for monocular 3D human pose estimation, our input remains the same as conventional lifting-based methods: a 2D pose sequence. Such 2D joint coordinates undoubtedly cause visual representation reduction compared to raw images. This inherent information entropy of the 2d pose sequence limits the theoretical upper bound on performance."
>
> **We would like to delve deeper into possiable solution for this general limitation. Please refer to our response to Question 1.**
>
> ---
>
> >**Weakness 2**: The accuracy gain brought by the mutual information loss is **limited**, as shown in Table 4.
>
> The improvement brought by the mutual information loss is modest because the ablation study on MI loss in the original paper was conducted based on a shared bottom incorporating the multi-task branch and task-aware decoder, which approaching its performance saturation. To further validate the effectiveness of our proposed MI loss, we conduct a **new ablation study with a model configuration of "Shared Bottom + TAD + MI Loss."**  We do not employ the "Shared Bottom + MI Loss" model configuration because MI loss is specifically designed for Task-Aware Decoder (TAD). MI Loss can not be directly integrated into the shared bottom.
>
> |Model Setting|Shared Bottom|Task-Aware Decoder|Mutual Information|MPJPE|
> |--|--|--|--|--|
> |Shared Bottom|√|-|-|50.6|
> |+ TAD Only|√|√|-|46.2|
> |+ TAD + MI Loss **(New)**|√|√|√|44.0|
>
> **As shown in the table, our MI loss improves the performance of the TAD module, reducing MPJPE by 2.2mm from 46.2mm to 44.0mm.** This result further demonstrates the effectiveness of our proposed MI loss. We have incorporated this result into Table 4. Please check our updated manuscript.
>
> ---
>
> >**Weakness 3**: The feature distribution visualization in Figure 6 reveals the disentangled states of pose and depth features within their feature space. However, since the 2D XY coordinates and the Z coordinate of the 3D pose have different value distributions, it is natural to see their original features distributed across each other. **Some comparisons with other lifting-based methods may be needed.**
>
> We have added the visualization of three feature distributions including the 2D pose feature (PrML), per-joint depth feature (PrML), and entangled feature from MotionBERT(Lifting-Based) in our Appendix.F "Additional Visualization".  Visualization results show that the distribution of entangled features is more compact, while our 2D pose features and per-joint depth features exhibit a more diverse distribution, indicating the potential to learn various representations. **Please check Appendix.F "Additional Visualization" (pages 23-25) in our updated paper for more details.**
>
> ---

---

> ### Author Response · Authors · 2024-11-23
> **Authors Response to Reviewer #AWXr (Part 2/3)**
>
> >**Question 1**: Did the authors **attempt to use image features** as input to improve the depth prediction capability, given that the lifting problem is still ill-posed?
>
> As the inherent information entropy of the input 2D pose sequence, the performance of methods only using 2D pose sequences as input gradually approaches the theoretical limit. Incorporating image features is a potential solution to this general limitation. However, there are two main challenges associated with using image features:
>
> 1. How to effectively **fuse** image features.
>
> There are several single-frame lifting-based methods that utilize image features, such as ContextPose [1], Lifting by Image [2], and CA-PF [3]. Unfortunately, integrating these methods into existing multi-frame lifting methods or our PrML is challenging due to their complex model architectures. **Therefore, we employ one cross-attention layer to fuse image features, making a preliminary attempt to fuse image features into our framework.**
>
> 2. How to utilize **multi-frame** image features.
>
> Due to the lightweight nature of 2D poses, traditional lifting-based frameworks typically input a 2D pose sequence. However, inputting a long-term image sequence is clearly impractical. How to effectively extend image input from a single frame to multiple frames remains an open question. To tackle this problem, we employ VAE to compress the original images into latent features, allowing for multi-frame input. We use the pre-trained VAE [4] from SDXL [5]. Although we have compressed the images using VAE, processing 243 frames remains computationally expensive. **Therefore, we leverage VAE to compress 15 frames image feature, making a preliminary attempt to fuse multi-frame image features into our framework.**
>
>
> |Method|T|MPJPE|
> |--|--|--|
> |PrML|15| 49.7 |
> |+ Image Feature for 2D|15|47.2|
> |+ Image Feature for Depth|15|49.6|
> |+ Image Feature for All|15|49.5|
>
> As shown in the table, despite using simple cross-attention to fuse image features, we still observe performance improvement. When we incorporated image features into the 2D branch, we observed a significant improvement. This is likely due to the fact that image features can easily capture 2D spatial information. However, when we added image features to the depth branch, the improvement was limited. This is because we used a simple cross-attention mechanism to process the images, and extracting depth information from RGB images is a challenging task. Similarly, when we incorporated image features into all branches, the improvement was modest. These experiments demonstrate the potential for further extending our framework. **We added this experiment results in our Appendix.C.6 "Attempt to Use Image Feature". (pages 18-19)**
>
>
> Moreover, the insights from multi-task learning can be further extended. Using 2D pose detectors to estimate 2D poses has been widely used, so **why not similarly utilize powerful depth estimation networks (like DepthAnything [6]) to estimate a relative depth from the image to facilitate 3D human pose estimation as well?** Due to time constraints, it is hard to design a novel model from scratch during the rebuttal phase. However, we can integrate depth features into existing single-frame methods that utilize image features to validate this idea. As shown in the table, by incorporating depth features from DepthAnything, we can reduce the error of CA-PF by 1.9mm, which demonstrates the effectiveness of our idea.
>
> |Method|MPJPE|
> |--|--|
> |CA-PF|41.4|
> |+ Depth Feature|39.5|
>
>
>
> In summary, how to leverage image features to facilitate multi-frame 3D human pose estimation is a non-trivial question and **our future research direction.** Your insightful question and our discussion will undoubtedly bring new insights to the monocular 3D human pose estimation community and inspire more research to think outside of the conventional lifting-based framework.
>
> [1] Context Modeling in 3D Human Pose Estimation: A Unified Perspective. CVPR'21
>
> [2] Lifting by Image - Leveraging Image Cues for Accurate 3D Human Pose Estimation. AAAI'24
>
> [3] A Single 2D Pose with Context is Worth Hundreds for 3D Human Pose Estimation. NeurIPS'24
>
> [4] Auto-Encoding Variational Bayes. ICLR'14
>
> [5] SDXL: Improving Latent Diffusion Models for High-Resolution Image Synthesis
>
> [6] Depth Anything: Unleashing the Power of Large-Scale Unlabeled Data. CVPR'24
>
> ---

---

> ### Author Response · Authors · 2024-11-23
> **Authors Response to Reviewer #AWXr (Part 3/3)**
>
> >**Question 2**: What is the **meaning of the statement regarding the different initial states of the 2D pose and per-joint depth?** Does this indicate an error in the detected 2D pose?
>
> Sorry for the confusion, we're here for further clarification. **The initial state of the 2D pose is well-detected, while the initial state of per-joint depth is unknown.** Specifically, given a 2D pose detected in image coordinates by a 2D pose detector, our goal is to estimate its corresponding 3D pose in the camera coordinates. The 2D pose part of the 3D pose is built upon the well-detected 2D pose. We need to learn a coordinate transformation to project it from image coordinates to camera coordinates. In contrast, the initial state of the per-joint depth in the 3D pose is totally unknown. We need to learn the per-joint depth from scratch, which is challenging and ambiguous.
>
> ---
>
> >**Question 3**: The qualitative examples of the inaccurate projected 2D poses in Figure 2 may be insufficient. **Some quantitative metrics comparing the accuracy of the projected 2D poses*** versus existing methods could provide stronger evidence.
>
> **We report the PCKh@0.5 following MPII [1] to evaluate the projected 2D pose.** We do not report the Average Precision (AP) because the calculation of AP requires the calculation of Object Keypoint Similarity (OKS), and the calculation of OKS requires dataset-specific standard deviations, which are not provided in Human3.6M. For more details, please refer to: https://cocodataset.org/#keypoints-eval (1.2. Object Keypoint Similarity and 1.3. Tuning OKS)
>
> |Method|PCKh@0.5|
> |--|--|
> |MotionBERT|95.9|
> |PrML|96.4|
>
> The table shows that our PrML achieves better accuracy in the projected 2D poses than the MotionBERT. We added this quantitative comparison in our Appendix.F "Additional Visualization". (page 20)
>
> Moreover, we have conducted a quantitative analysis of Mean Per Joint Position Error (MPJPE) across different axes for all actions and three hard actions [1], with the results presented in Figure 3 of our original paper. **This includes a quantitative comparison of the accuracy of the 2D pose (X-Y axes) before projecting them to image coordinates.** Please refer to Figure 3 of our paper for more details. Alternatively, please refer to the following table:
>
> |MPJPE (2D part)|All actions|Photo|Sitting|Sitting Down|
> |--|--|--|--|--|
> |MotionBERT|18.6|22.8|23.9|27.3|
> |PrML|17.9|20.9|22.8|25.4|
>
> The table shows that our method achieves the best average performance across all actions compared with MotionBERT, especially on challenging actions [1].
>
> [1] 2D Human Pose Estimation: New Benchmark and State of the Art Analysis. CVPR'14
>
> [2] Learning Skeletal Graph Neural Networks for Hard 3D Pose Estimation. ICCV'21
>
> ---
>
> Thanks again for your valuable time and considerate advice, which help us improve the quality of our work. We hope our effort could address your concerns. We are more than willing to provide further clarifications if there are any lingering questions or concerns.
>
> Best!
>
> The authors of Paper #2366

---

> > ### Author Response · Authors · 2024-11-25
> > **Please let us know whether all the issues are addressed**
> >
> > Dear reviewer,
> >
> > Thanks for the comments. We have provided more explanations and results to your questions. Please let us know whether we have addressed all the issues as the deadline (Nov 26) is fast approaching.
> >
> > Please also consider raising the score after all the issues are addressed.
> >
> > Thank you,

---

> ### Comment · Reviewer_AWXr · 2024-11-25
> **Response to the authors**
>
> Thanks for the response from the authors. The newly released results have addressed some concerns, except for the visualization of the feature distribution. The entangled features of MotionBERT are also presented in the figures with 2D pose features and depth features. These distributions are still confusing for me in understanding their differences. Putting the features from different methods in a shared feature space may not provide strong evidence to show the advantages.

---

> > ### Author Response · Authors · 2024-11-25
> > **Authors Response to Reviewer #AWXr**
> >
> > Dear Reviwer #AWXr,
> >
> > Thanks for your valuable time in reviewing our rebuttal.
> >
> > ---
> >
> > >These distributions are still confusing for me in understanding their differences. Putting the features from different methods in a shared feature space may not provide strong evidence to show the advantages.
> >
> > Sorry for the confusion, we're here for further clarification. As shown in Figures 13, 14, and 15, the 2D pose features, depth features, and entangled features are clearly distinguishable into three distinct clusters. It is the different distributions of these three features that allow t-SNE to classify them so distinctly.
> >
> > To address your concerns further, we have added more results that visualize the feature distribution of their own feature space. The figure shows that our method forms clusters for different joints, capturing their relative positions. However, entangled features lack this ability and cause overlap and redundancy. **Please check Appendix.F "Additional Visualization" (page 26) in our updated paper for more details.**
> >
> > ---
> >
> > Thanks again for your valuable time. We hope our effort could address your concerns. We are more than willing to provide further clarifications if there are any lingering questions or concerns.
> >
> > Best!
> >
> > The authors of Paper #2366

---

> > ### Author Response · Authors · 2024-12-02
> >
> > Dear Reviwer #AWXr,
> >
> > Thanks for your valuable time in reviewing our rebuttal. Please let us know whether we have addressed the concern about feature distribution visualization as the extended deadline (December 2nd) is fast approaching. Additionally, please allow us to reaffirm the **novelty** and **contribution** of our work.
> >
> > ---
> >
> > ### 1. We Identify and Tackle a **New Research Problem**
> >
> > Since SimpleBaseline[1] proposed the lifting-based framework, most subsequent works [2-9] have followed this framework and have dominated the monocular 3D human pose estimation field. However, these works mainly focus on developing various encoders within the lifting framework, **without stepping outside of this framework to explore the impact of the lifting process itself for 3D human pose estimation.** We point out in this paper with quantitative and qualitative evidence that the lifting-based framework, encoding the well-detected 2D pose features and the unknown per-joint depth features in an entangled feature space, will inevitably introduce uncertainty to the 2D pose and cause erosion. How to address the **erosion of well-detected 2D pose caused by depth uncertainty** arising from entangled feature space is a non-trivial problem.
> >
> > ---
> >
> > ### 2. We Provide **Fresh Insight** regarding the Problem Cause
> >
> > We discover the fundamental cause of the erosion of well-detected 2D pose caused by depth uncertainty is **encoding them in an entangled feature space.** In light of this, we provide a straightforward yet powerful insight to solve this problem: **introducing multi-task learning to estimate 2D pose and per-joint depth separately.** This insight is fresh in the context of monocular 3D human pose estimation.
> >
> > ---
> >
> > ### 3. We Design a **Novel Framework** to Solve the Problem
> >
> > How to address the erosion of well-detected 2D pose caused by depth uncertainty arising from entangled feature space is a non-trivial problem. We comprehensively consider the task pipeline and design a novel **progressive multi-task learning framework named PrML.** The first step of PrML introduces two task branches: refining the well-detected 2D pose features and learning the per-joint depth features. The second step of PrML employs a task-aware decoder to indirectly supplement the complementary information between the refined 2D pose features and the learned per-joint depth features. Extensive experiments on two widely used monocular 3D human pose estimation benchmarks (i.e., Human3.6M and MPI-INF-3DHP) demonstrate that the **proposed progressive multi-task learning framework outperforms conventional lifting-based framework in terms of accuracy and robustness with fewer parameters.**
> >
> > ---
> >
> > ### 4. Contribution to Community
> >
> > As mentioned in our future work,  It will be novel and interesting to design specific encoders for different tasks to extend our framework in future research.
> >
> > Moreover, the insights from multi-task learning can be further extended. First, we perform an embarrassingly simple transformation: replacing the original single regression head of the lifting-based framework with two regression heads to expand our insight into previous lifting-based methods. Such simple modification enables MixSTE[4] (CVPR'22) to outperform the latest method KTPFormer[9] (CVPR'24). Second, using 2D pose detectors to estimate 2D poses has been widely used, so why not similarly utilize powerful depth estimation networks (like DepthAnything[10]) to estimate a relative depth from the image to facilitate 3D human pose estimation as well?
> >
> > We provide all code and documents for the community and hope our work inspires more research to think outside of the conventional lifting-based framework. We also believe that a research community should embrace different frameworks, and our work can be such a starting point.
> >
> >
> > ---
> >
> > Thanks again for your valuable time. We hope our effort could address your concerns. Please also consider raising the score after all the issues are addressed.
> >
> > Best!
> >
> > The authors of Paper #2366
> >
> >
> > [1] A simple yet effective baseline for 3d human pose estimation. ICCV'17
> >
> > [2] 3D human pose estimation with spatial and temporal transformers. ICCV'21
> >
> > [3] MHFormer: Multi-hypothesis transformer for 3D human pose estimation. CVPR'22
> >
> > [4] MixSTE: Seq2seq Mixed Spatio-Temporal Encoder for 3D Human Pose Estimation in Video. CVPR'22
> >
> > [5] P-STMO: Pre-Trained Spatial Temporal Many-to-One Model for 3D Human Pose Estimation. ECCV'22
> >
> > [6] 3D Human Pose Estimation with Spatio-Temporal Criss-cross Attention. CVPR'23
> >
> > [7] PoseFormerV2: Exploring frequency domain for efficient and robust 3D human pose estimation. CVPR'23
> >
> > [8] MotionBERT: A unified perspective on learning human motion representations. ICCV'23
> >
> > [9] KTPFormer: Kinematics and Trajectory Prior Knowledge-Enhanced Transformer for 3D Human Pose Estimation. CVPR'24
> >
> > [10] Depth Anything: Unleashing the Power of Large-Scale Unlabeled Data. CVPR'24

---

### Official Review · Reviewer_KwAb · 2024-10-31

**Soundness:** 2
**Presentation:** 3
**Contribution:** 3
**Rating:** 6
**Confidence:** 5

**Summary:**

The manuscript proposes a multi-task learning method that independently models and regresses 2D pose and depth features, achieving good performance on 3D HPE benchmarks. Its advantages and weakness are as follows:

Advantages:
1. The motivation of the proposed method is solid.
2. The manuscript is well written and clearly presented.
3. The multi-task learning method with two separate regression heads demonstrates effectiveness when applied to other approaches.

Weakness:
1. The difference between the descriptions/ theoretical analysis in the paper and the provided source code, particularly regarding the model architecture and the key proposal of mutual information loss.
2. The effectiveness of the mutual information loss requires further validation.
3. Lack of related comparison method.

**Strengths:**

1. The motivation of the proposed method is solid. 3D pose estimation lifted from detected 2D joints is inherently an ambiguous problem, since the output 2D pose has a well-detected 2D pose as initialization but the depth is learned from scratch. The authors suggest that the unknown depth may erode the 2D pose in the entangled feature space. To tackle this problem, they propose a PrML model with two separate branches to learn 2D pose and joint depth. On this basis, a task-aware decoder is developed to connect the 2D pose features and joint depth features. The decoded features are then regressed with two separate heads to obtain the 2D pose and depth coordinates.
2. The manuscript is well written. Combining the visualization analysis, the authors clearly present the rationale behind the motivation and demonstrate the model's improvements in relevant aspects.
3. The effectiveness of the multi-task learning concept is validated on other methods. Significant performance improvements are obtained by setting up two separate regression heads to independently optimize 2D and depth coordinates.

**Weaknesses:**

However, after reading the code provided by the authors, I have several questions, mostly about the difference between the code and the descriptions in the manuscript.
1) The architecture of shared bottom in PrML. From the code, the shared bottom follows the main architecture of MotionBert, of which spatial and temporal blocks are differently ordered to form two distinct branches, with the output features adaptively fused with an attention regressor. It is a typical architecture rather than the simple description of “Vanilla self-attention” in the manuscript. The authors should provide a more detailed and accurate description of the shared bottom architecture.
2) Difference in the Mutual Information Loss. In the code, the mutual information comprises two parts: a. the KL Divergence between pose label Y2D and pose support feature S2D; b. the KL Divergence between depth label YD and depth support feature SD. However, the descriptions in equations 5-7 which introduce the distribution dependency between support features (S2D/SD) and bias features (B2D/BD) are not implemented in the source code. The authors should address this inconsistency to improve the paper's clarity and reproducibility.
3) The impact of Mutual Information Loss. In the code, the KL Divergence losses for pose and depth are assigned small coefficients of 0.005 and 0.05, respectively, and are further multiplied with a 0.01 factor in the final loss function, resulting in a very small overall contribution. Thus, the impact of Mutual Information Loss should be further verified to demonstrate its importance as one of the key contributions in the paper.
4) The results on Human3.6M are based on 2D pose detected by SH, which is in consistency with MotionBERT. However, SH-detected 2D poses from Human3.6M are not widely used currently, as most lifting-based methods rely on CPN-detected poses. Therefore, the other competitor MotionAGFormer[1] which also utilizes the SH detected 2D poses should be presented for a more comprehensive comparison of properties.
[1] MotionAGFormer: Enhancing 3D Human Pose Estimation With a Transformer-GCNFormer Network, WACV2024

**Questions:**

Same with the weakness.

---

> ### Author Response · Authors · 2024-11-23
> **Authors Response to Reviewer #KwAb (Part 1/2)**
>
> Dear Reviwer #KwAb,
>
> Thanks for your valuable time in reviewing our work. We will address your concerns in the following part. The corresponding revised content is highlighted in red font within our updated paper.
>
> ---
>
> >**Weakness 1**: The architecture of shared bottom in PrML. From the code, the shared bottom follows the main architecture of MotionBert, of which spatial and temporal blocks are differently ordered to form two distinct branches, with the output features adaptively fused with an attention regressor. It is a typical architecture rather than the simple description of “Vanilla self-attention” in the manuscript. **The authors should provide a more detailed and accurate description of the shared bottom architecture.**
>
> We regret that our simplified content has sacrificed precise description. We refer to them as "vanilla self-attention" because both spatial and temporal blocks perform vanilla self-attention, differing in the dimensions over which they are computed. Specifically, the spatial block operates on tensors of shape (B, T, **J**, **C**), while the temporal block operates on tensors of shape (B, J, **T**, **C**), where B is the batch size, T is the number of frames, J is the number of joints, and C is the feature dimension. We summarize both blocks as "vanilla self-attention" in our original manuscript to simplify our description.
>
> We apologize for our description not being as accurate as it should have been. **According to your constructive feedback, we have enhanced the clarity and accuracy of our description of the shared bottom architecture in our updated manuscript.** The revised content is indicated in red. Please check our updated manuscript. Alternatively, you can directly examine the following content.
>
> Lines 253-257: "We first use a linear embedding layer to project the 2D pose sequence into high-dimensional features. Then, we employ the DSTFormer block proposed by MotionBERT (Zhu et al., 2023) as our shared bottom to extract general features $F \in \mathbb{R}^{T \times J \times C}$. The DSTFormer block is composed of spatial-temporal and temporal-spatial branches. The outputs of two branches are adaptively fused by an attention regressor."
>
> Lines 437-438: "We follow this design and use the DSTFormer block from MotionBERT (Zhu et al., 2023) as the shared bottom."
>
> ---
>
> >**Weakness 2**: Difference in the Mutual Information Loss. In the code, the mutual information comprises two parts: a. the KL Divergence between pose label Y2D and pose support feature S2D; b. the KL Divergence between depth label YD and depth support feature SD. However, the descriptions in equations 5-7 which introduce the distribution dependency between support features (S2D/SD) and bias features (B2D/BD) are not implemented in the source code. **The authors should address this inconsistency to improve the paper's clarity and reproducibility.**
>
> Our provided code is a simplified implementation. Taking the 2D pose optimization objective $\mathcal{I}(Y_{2D};S_{2D}) - \mathcal{I}(S_{2D};B_{2D})$ as an example. Since both $\mathcal{I}(Y_{2D};S_{2D})$ and $\mathcal{I}(S_{2D};B_{2D})$ are non-negarive (KL Divergence $\geq$ 0), the $\mathcal{I}(Y_{2D};S_{2D}) - \mathcal{I}(S_{2D};B_{2D})$ will result in negative values during training. This will cause the training process to fail to converge. Therefore, we simplified the implementation of mutual information by calculating only the first term.
>
> We apologize for the lack of implementation details regarding mutual information loss which introduces inconsistency. **According to your practical suggestion, we have added the above implementation details of our mutual information loss in Appendix.E "Additional Analysis of Mutual Information" in our updated manuscript.** (page 20)
>
> ---

---

> ### Author Response · Authors · 2024-11-23
> **Authors Response to Reviewer #KwAb (Part 2/2)**
>
> >**Weakness 3**: The impact of Mutual Information Loss. In the code, the KL Divergence losses for pose and depth are assigned small coefficients of 0.005 and 0.05, respectively, and are further multiplied with a 0.01 factor in the final loss function, resulting in a very small overall contribution. **Thus, the impact of Mutual Information Loss should be further verified to demonstrate its importance as one of the key contributions in the paper.**
>
> **The first two coefficients (0.005 and 0.05) act as scaling factors for the KL divergence losses, while the final coefficient of 0.01 represents the overall contribution of the MI loss to the total loss.** The KL divergence losses for 2D pose and depth yielded huge values. The 2D pose KL loss usually ranged from 10 to 100, while the depth KL loss was generally between 1 to 10. In contrast, the 3D pose loss is generally bounded ranging from 0 to 0.5. The coefficients of 0.005 for 2D pose and 0.05 for depth are applied to scale down the KL divergence loss to the range of [0, 0.5], preventing large loss values that could hinder training convergence. The coefficients 0.005 and 0.05 correspond to $\lambda_{2D}$ and $\lambda_{D}$ in Equation 7 of our paper, respectively. The coefficient 0.01 in our final loss function corresponds to $\lambda_{MI}$ in Equation 9 of our paper, which measures the overall contribution. Ablation studies were conducted to evaluate the effect of this parameter as:
>
> |$\lambda_{MI}$|MPJPE|
> |--|--|
> |0|38.6|
> |1|38.6|
> |0.1|38.5|
> |0.01|38.2|
>
> This result has been appended to our  Appendix C.4 "Analysis on Hyperparameter setting". Please check our updated manuscript.
>
> Moreover, the ablation study on MI loss in the original paper was conducted based on a shared bottom incorporating the multi-task branch and task-aware decoder. As a result, the improvement was relatively modest, primarily due to the performance approaching its theoretical limit. To further validate the effectiveness of our proposed MI loss, we added a **new ablation study with a model configuration of "Shared Bottom + TAD + MI Loss."**  We do not employ the "Shared Bottom + MI Loss" model configuration because MI loss is specifically designed for Task-Aware Decoder (TAD). MI Loss can not be directly integrated into the shared bottom.
>
> |Model Setting|Shared Bottom|Task-Aware Decoder|Mutual Information|MPJPE|
> |--|--|--|--|--|
> |Shared Bottom|√|-|-|50.6|
> |+ TAD Only|√|√|-|46.2|
> |+ TAD + MI Loss **(New)**|√|√|√|44.0|
>
>
> As shown in the table, our MI loss achieved an 2.2 mm error drop (from 46.2mm to 44.0mm), demonstrating its effectiveness. We have incorporated this result into Table 4. Please check our updated manuscript.
>
> ---
>
> >**Weakness 4**: The results on Human3.6M are based on the 2D pose detected by SH, which is in consistency with MotionBERT. However, SH-detected 2D poses from Human3.6M are not widely used currently, as most lifting-based methods rely on CPN-detected poses. **Therefore, the other competitor MotionAGFormer[1] which also utilizes the SH detected 2D poses should be presented for a more comprehensive comparison of properties.** [1] MotionAGFormer: Enhancing 3D Human Pose Estimation With a Transformer-GCNFormer Network, WACV2024
>
> Thanks for sharing such great work, MotionAGFormer[1]. We discuss this paper in our Related Work section in our updated manuscript. Please check our updated manuscript. Alternatively, you can directly examine the following content.
>
> Lines 163-165: ". Moreover, MotionAGFormer (Soroush Mehraban, 2024) using two parallel transformer and GCNFormer streams to better learn the underlying 3D structure."
>
> We also incorporate MotionAGFormer's impressive performance in Tables 1, 2, and 3 in the Experiments section and Table 12 in Appendix.C.5 "Additional Quantitative Results". Additionally, we find other related work[2] that further explores the lifting-based 3D human pose estimation. We have cited it in our updated paper (line 194).
>
> [1] MotionAGFormer: Enhancing 3D Human Pose Estimation With a Transformer-GCNFormer Network. WACV'2024
>
> [2] Evaluating Recent 2D Human Pose Estimators for 2D-3D Pose Lifting.
>
>
> ---
>
> Thanks again for your valuable time and considerate advice, which help us improve the quality of our work. We hope our effort could address your concerns. We are more than willing to provide further clarifications if there are any lingering questions or concerns.
>
> Best!
>
> The authors of Paper #2366

---

> > ### Author Response · Authors · 2024-11-25
> > **Please let us know whether all the issues ared addressed**
> >
> > Dear reviewer,
> >
> > Thanks for the comments. We have provided more explanations and results to your questions. Please let us know whether we have addressed all the issues as the deadline (Nov 26) is fast approaching.
> >
> > Please also consider raising the score after all the issues are addressed.
> >
> > Thank you,

---

> > > ### Comment · Reviewer_KwAb · 2024-11-25
> > >
> > > My concerns regarding 2. Difference in the Mutual Information Loss remain unresolved. While the definition of Mutual Information Loss is clearly stated in both the main manuscript and the appendix, the noticeable gap between the presented equations and the specific code implementation cannot simply be explained with "preventing training from converging." Ensuring consistency between the theory presentation and code implementation is essential for maintaining responsibility for the community.

---

> > > > ### Author Response · Authors · 2024-11-26
> > > >
> > > > Dear Reviwer #KwAb,
> > > >
> > > > Thanks for your valuable time in reviewing our rebuttal.
> > > >
> > > > ---
> > > >
> > > > >My concerns regarding 2. Difference in the Mutual Information Loss remain unresolved. While the definition of Mutual Information Loss is clearly stated in both the main manuscript and the appendix, the noticeable gap between the presented equations and the specific code implementation cannot simply be explained with "preventing training from converging." Ensuring consistency between the theory presentation and code implementation is essential for maintaining responsibility for the community.
> > > >
> > > > We agree with you that we should maintain responsibility for the community. We also fully understand your concerns about reproducibility. **This responsibility for the community is precisely why we provide reviewers with our all code, a detailed README, and checkpoints for review, regardless of whether our work is accepted.**
> > > >
> > > > Specifically, term $\mathcal{I}(Y_{2D};S_{2D}) - \mathcal{I}(S_{2D};B_{2D})$ will yield negative values during training. A negative loss value will lead to the vanishing gradients problem and prevent training from converging. This situation is **fatal** for deep learning methods. Therefore, we simplified the implementation of mutual information by calculating only the first term $\mathcal{I}(Y_{2D};S_{2D})$ to guarantee the value of our loss is non-negative. This simplified implementation achieve satisfactory results.
> > > >
> > > > To address your concerns further, we **updated the corresponding content both in the main manuscript and the appendix to ensure consistency with our final code implementation.** Please check our updated manuscript. Alternatively, you can directly examine the following content.
> > > >
> > > > Lines 306-309: "Since both $\mathcal{I}(Y_{D};S_{D})$ and $\mathcal{I}(S_{D};B_{D})$ are non-negative, the $\mathcal{I}(Y_{D};S_{D}) - \mathcal{I}(S_{D};B_{D})$ will yield negative values during training, leading to vanishing gradients problem and preventing training from converging. Therefore, we simplified the implementation of mutual information by calculating only the first term $\mathcal{I}(Y_{D};S_{D})$."
> > > >
> > > > Equation 7:  $\mathcal{L} _ {MI} = \lambda_{2D}\mathcal{I}(Y_{2D};S_{2D}) + \lambda_{D}\mathcal{I}(Y_{D};S_{D})$
> > > >
> > > > Equation 12 (Appendix):  "Our final optimization objective becomes: $\textbf{max  }\mathcal{I}({Y};{S}\mid {B}) \xrightarrow[]{\quad} \textbf{max  } \mathcal{I}({Y};{S})$"
> > > >
> > > > ---
> > > >
> > > > Thanks again for your valuable time. We hope our effort could address your concerns. We are more than willing to provide further clarifications if there are any lingering questions or concerns.
> > > >
> > > > Best!
> > > >
> > > > The authors of Paper #2366

---

> > > > ### Author Response · Authors · 2024-12-02
> > > >
> > > > Dear Reviwer #KwAb,
> > > >
> > > > Thanks for your valuable time in reviewing our rebuttal. Please let us know whether we have addressed the concern about mutual information loss as the extended deadline (December 2nd) is fast approaching. Additionally, please allow us to reaffirm the **novelty** and **contribution** of our work.
> > > >
> > > > ---
> > > > ### 1. We Identify and Tackle a **New Research Problem**
> > > >
> > > > Since SimpleBaseline[1] proposed the lifting-based framework, most subsequent works [2-9] have followed this framework and have dominated the monocular 3D human pose estimation field. However, these works mainly focus on developing various encoders within the lifting framework, **without stepping outside of this framework to explore the impact of the lifting process itself for 3D human pose estimation.** We point out in this paper with quantitative and qualitative evidence that the lifting-based framework, encoding the well-detected 2D pose features and the unknown per-joint depth features in an entangled feature space, will inevitably introduce uncertainty to the 2D pose and cause erosion. How to address the **erosion of well-detected 2D pose caused by depth uncertainty** arising from entangled feature space is a non-trivial problem.
> > > >
> > > > ---
> > > >
> > > > ### 2. We Provide **Fresh Insight** regarding the Problem Cause
> > > >
> > > > We discover the fundamental cause of the erosion of well-detected 2D pose caused by depth uncertainty is **encoding them in an entangled feature space.** In light of this, we provide a straightforward yet powerful insight to solve this problem: **introducing multi-task learning to estimate 2D pose and per-joint depth separately.** This insight is fresh in the context of monocular 3D human pose estimation.
> > > >
> > > > ---
> > > >
> > > > ### 3. We Design a **Novel Framework** to Solve the Problem
> > > >
> > > > How to address the erosion of well-detected 2D pose caused by depth uncertainty arising from entangled feature space is a non-trivial problem. We comprehensively consider the task pipeline and design a novel **progressive multi-task learning framework named PrML.** The first step of PrML introduces two task branches: refining the well-detected 2D pose features and learning the per-joint depth features. The second step of PrML employs a task-aware decoder to indirectly supplement the complementary information between the refined 2D pose features and the learned per-joint depth features. Extensive experiments on two widely used monocular 3D human pose estimation benchmarks (i.e., Human3.6M and MPI-INF-3DHP) demonstrate that the **proposed progressive multi-task learning framework outperforms conventional lifting-based framework in terms of accuracy and robustness with fewer parameters.**
> > > >
> > > > ---
> > > >
> > > > ### 4. Contribution to Community
> > > >
> > > > As mentioned in our future work,  It will be novel and interesting to design specific encoders for different tasks to extend our framework in future research.
> > > >
> > > > Moreover, the insights from multi-task learning can be further extended. First, we perform an embarrassingly simple transformation: replacing the original single regression head of the lifting-based framework with two regression heads to expand our insight into previous lifting-based methods. Such simple modification enables MixSTE[4] (CVPR'22) to outperform the latest method KTPFormer[9] (CVPR'24). Second, using 2D pose detectors to estimate 2D poses has been widely used, so why not similarly utilize powerful depth estimation networks (like DepthAnything[10]) to estimate a relative depth from the image to facilitate 3D human pose estimation as well?
> > > >
> > > > We provide all code and documents for the community and hope our work inspires more research to think outside of the conventional lifting-based framework. We also believe that a research community should embrace different frameworks, and our work can be such a starting point.
> > > >
> > > > ---
> > > >
> > > > Thanks again for your valuable time. We hope our effort could address your concerns. Please also consider raising the score after all the issues are addressed.
> > > >
> > > > Best!
> > > >
> > > > The authors of Paper #2366
> > > >
> > > > [1] A simple yet effective baseline for 3d human pose estimation. ICCV'17
> > > >
> > > > [2] 3D human pose estimation with spatial and temporal transformers. ICCV'21
> > > >
> > > > [3] MHFormer: Multi-hypothesis transformer for 3D human pose estimation. CVPR'22
> > > >
> > > > [4] MixSTE: Seq2seq Mixed Spatio-Temporal Encoder for 3D Human Pose Estimation in Video. CVPR'22
> > > >
> > > > [5] P-STMO: Pre-Trained Spatial Temporal Many-to-One Model for 3D Human Pose Estimation. ECCV'22
> > > >
> > > > [6] 3D Human Pose Estimation with Spatio-Temporal Criss-cross Attention. CVPR'23
> > > >
> > > > [7] PoseFormerV2: Exploring frequency domain for efficient and robust 3D human pose estimation. CVPR'23
> > > >
> > > > [8] MotionBERT: A unified perspective on learning human motion representations. ICCV'23
> > > >
> > > > [9] KTPFormer: Kinematics and Trajectory Prior Knowledge-Enhanced Transformer for 3D Human Pose Estimation. CVPR'24
> > > >
> > > > [10] Depth Anything: Unleashing the Power of Large-Scale Unlabeled Data. CVPR'24

---

> > > > > ### Comment · Reviewer_KwAb · 2024-12-03
> > > > >
> > > > > 1.
> > > > > I have read the revised manuscript, and noted the authors' supplemented description about Mutual Information Loss. I think it solved the discrepancy between the code and the manuscript. However, this makes the formulation process of Mutual Information Loss appears overly complex and somewhat redundant. In other words, if the objective is to improve the mutual information between support feature and label, why construct a more complicate formulation with unrelated features or variants?
> > > > >
> > > > > I am not sure if the solution is appropriate, and will leave this issue for AC’s consideration.
> > > > >
> > > > > 2.
> > > > > Overall, the most effective proposal of the PrML appears to be the multitask learning method which regresses 2D pose and depth separately. It achieves significant performance increase when applied to other lifting-based methods.
> > > > >
> > > > > However, the novelty and the superiority of the model itself seems incremental. For example, the MotionAGFormer shows comparable performance compared with PrML in Table 1. If MotionAGFormer were equipped with two separate regression heads as other competitors in Table 7, it is likely to achieve even better accuracy based on the observed MPJPE enhancement. Meanwhile, I still hold the view that the impact of Mutual Information Loss is marginal.
> > > > >
> > > > > 3.
> > > > > After reviewing other reviewers’ comments, I quiet agree that the baseline performance in the Ablation Study appears too weak. This undermines the persuasiveness of the proposed modules' effectiveness.
> > > > >
> > > > > I think the proposed method has good aspects and strength, as highlighted in my initial comments. But it also has some deficiency. Therefore, I will keep my rate as 6, and leave the decision to AC.

---

> ### Author Response · Authors · 2024-12-03
>
> Dear Reviwer #KwAb,
>
> Thanks for your valuable time in reviewing our rebuttal. We sincerely appreciate your positive attitude towards our work. We would like to further clarify our work in the following part.
>
> ---
>
> ### **1.Baseline**
>
> We present a progressive multi-task learning framework that offers a **new paradigm for monocular 3D human pose estimation.** The performance of the shared bottom is lower than the previous state-of-the-art method can be attributed to the relatively **small feature dimension of 128.** The shared bottom is used to learn a basic feature representation. In contrast, previous lifting methods typically set the **feature dimension to 512.**
>
> If we add an additional multi-task branch to the previous model, it will undoubtedly lead to an improvement due to the increased number of parameters. Therefore, we start ablation studies with a simple shared bottom as a baseline to construct our PrML step by step.
>
> Actually, experiments on Human3.6M and MPI-INF-3DHP demonstrate that the **proposed progressive multi-task learning framework outperforms conventional lifting-based framework in terms of accuracy and robustness with fewer parameters.** These experiments are enough to serve as an ablation study supporting the effectiveness of our multi-task learning insight.
>
>
> ### **2.Factual Errors**
>
> We would list some factual errors and arbitrary judgments in other reviewer's comments.
>
> > As stated, the benchmarks are introduced more than 10 years ago, and all of them are not from in-the-wild environments.
>
> MPI-INF-3DHP[1] dataset, contrary to the reviewer's assertion, is published in 2017 at 3DV. MPI-INF-3DHP[1] is a stander dataset for in-the-wild 3D human pose estimation, as evident from its title: "Monocular 3D Human Pose Estimation **In The Wild** Using Improved CNN Supervision."
>
> >By using their 2.5D representation, many methods, including 3DCrowdNet (CVPR 2022) successfully preserved well-detected input 2D pose (they even made the input 2D pose better by refining it with image features).
>
> 3DCrowdNet[3] using 2D pose heatmap and image feature, rather than 2.5D representation[2]. **3DCrowdNet[3] even do not cite [2]!**
>
>
> [1] Monocular 3D Human Pose Estimation In The Wild Using Improved CNN Supervision. 3DV'17
>
> [2] Hand Pose Estimation via Latent 2.5D Heatmap Regression. ECCV'18
>
> [3] Learning to estimate robust 3d human mesh from in-the-wild crowded scenes. CVPR'22
>
>
> >I’m not sure the 1 mm improvement is meaningful.
>
> >I know that Table 7 shows that the proposed item could be used in other methods as well. But still, 1 mm improvements are marginal.
>
> Table 7 provides a very simple alternative of how our multi-task learning concept can be applied to lifting methods. **Our rebuttal provides quantitative evidence** demonstrating that the 1mm improvement is worthwhile and enables MixSTE (CVPR'22) to outperform KTPFormer (CVPR'24).
>
> However, there is a reviewer who insists that this improvement is marginal **without objective evidence.** It's also interesting that although the reviewer states, "**I'm not sure** the 1 mm improvement is meaningful," the reviewer's confidence rate is "5: You are absolutely certain about your assessment."
>
> In summary, we strongly believe that a professional and valuable discussion should be conducted in a **"claim-evidence"** manner.
>
> ---
>
> Thanks again for your valuable time and positive attitude towards our work.
>
> Best!
>
> The authors of Paper #2366

---

### Author Response · Authors · 2024-11-23
**General Response**

We thank the reviewers for their valuable time and insightful feedback.

We are encouraged that reviews agree that our **motivation is solid**("The motivation of the proposed method is solid." (#KwAb)) and our paper is **well-written**("The paper is well-written and easy to follow." (#AWXr), "The manuscript is well written." (#KwAb)). We are further glad that reviewers agree that our work is **novel**("The paper proposes a novel perspective." (#AWXr)) and present **strong experiment results**("The experiments and demonstrations are thorough and persuasive." (#AWXr), "Authors clearly present the rationale behind the motivation and demonstrate the model's improvements in relevant aspects."(#KwAb)).

We appreciate all the suggestions made by reviewers to improve our work and will answer specific questions for each reviewer in the corresponding rebuttal space. We are looking forward to further feedback.

---

### Meta-Review · Area_Chair_x4CM · 2024-12-20

**Metareview:**

Post rebuttal, this work received two borderline accept and a reject. A reviewer who were positive also had some issues with the paper. One reviewer flagged the gap between the given loss equation and the implementation which is clearly a major issue. Also the other reviewer had issues with visualization of feature distribution.

The third reviewer felt that the framework-level modifications were considered neither novel nor complete, with limited experimental demonstrations that relied on outdated benchmarks and lacked in-the-wild evaluations. Additionally, the improvements reported were deemed marginal, as the baseline used was weaker than any method in the comparison tables, potentially exaggerating the contribution. Despite the rebuttal, these concerns remained unresolved, particularly regarding the limited novelty, weak baseline, and marginal performance gains (e.g., a 1mm improvement in Table 7).

After careful consideration, the Area Chair panel has decided not to accept the paper in its current form.

**Additional Comments On Reviewer Discussion:**

All the reviewers had some issues with the paper and the rebuttal and the discussion did not help the reviewers to raise the scores.

---

### Decision · Program_Chairs · 2025-01-22

Reject